# Enabling High Performance Cloud Computing for the Community Multiscale Air Quality Model (CMAQ) version 5.3.3: Performance Evaluation and Benefits for the User Community

Christos I. Efstathiou[1†], Elizabeth Adams[1], Carlie J. Coats[1], Robert Zelt[2], Mark Reed[2], John McGee[2], Kristen M. Foley[3], Fahim I. Sidi[3], David C. Wong [3], Steven Fine[4], Saravanan Arunachalam[1*]

[1]Institute for the Environment, The University of North Carolina at Chapel Hill, Chapel Hill, 27599, USA
[2]Research Computing, Information Technology Services, The University of North Carolina at Chapel Hill, Chapel Hill, 27599, USA
[3]Center for Environmental Measurement and Modeling, Office of Research and Development, U.S. Environmental Protection Agency, Research Triangle Park, NC, USA
[4]Formerly with the Office of Air and Radiation, U.S. Environmental Protection Agency, Washington, DC, USA

[†]Currently with Physicians, Scientist, and Engineers for Healthy Energy, Oakland, CA, USA

*Correspondence to*: Saravanan Arunachalam (sarav@email.unc.edu)

**Abstract.** The Community Multiscale Air Quality (CMAQ) Model is a local-to-hemispheric scale numerical air quality modeling system developed by the U.S. Environmental Protection Agency (USEPA) and supported by the Center for Community Modeling and Analysis System (CMAS). CMAQ is used for regulatory purposes by the USEPA program offices and state and local air agencies, and is also widely used by the broader global research community to simulate and understand complex air quality processes and for computational environmental fate and transport, and climate and health impact studies. Leveraging state-of-the-science cloud computing resources for high performance computing (HPC) applications, CMAQ is now available as a fully tested, publicly available technology stack (HPC cluster and software stack) for two major cloud service providers (CSPs). Specifically, CMAQ configurations and supporting materials have been developed for use on their HPC clusters, including extensive online documentation, tutorials, and guidelines to scale and optimize air quality simulations using their services. These resources allow modelers to rapidly bring together CMAQ, cloud-hosted datasets, and visualization and evaluation tools on ephemeral clusters that can be deployed quickly and reliably worldwide. Described here are considerations in CMAQ v5.3.3 cloud use and the supported resources for each CSP, presented through a benchmark application suite that was developed as an example of typical simulation for testing and verifying components of the modeling system. The outcomes of this effort are to provide findings from performing CMAQ simulations on the cloud using popular vendor provided resources, to enable the user community to adapt this for their own needs and identify specific areas of potential optimization with respect to storage and compute architectures.

## 1 Introduction

Over the past decade, cloud computing has received a tremendous amount of attention for its potential to enable and simplify high performance computing (HPC) applications. Modeling user communities can greatly benefit by having real time access to cloud-ready reproducible workflows that include complex models and large datasets. Benefits can include reduced effort required to manage computational resources, the ability to rapidly obtain more resources when needed, flexible approaches for managing costs, and new opportunities for convenient data sharing. State-of-the-science numerical models simulating a variety of different processes and scales ranging from global circulation models to regional and high-resolution weather prediction workloads, have been demonstrated to perform efficiently on HPC infrastructure in the cloud. Development groups for earth system models such as weather, climate, ocean circulation, and air quality are currently designing and deploying modeling platforms or components that utilize different cloud environments (Campbell et al., 2023; Powers et al., 2021; Zhuang et al., 2020; Chui et al., 2019; Eastham et al., 2018; Chen et al., 2017).

The vast majority of such applications leverage “Infrastructure as a Code” (IaaC) or “Infrastructure as a Service” (IaaS) technologies and storage options provided by different cloud service providers which creates the need for a flexible approach in terms of data integration. In the context of air quality models, cloud computing encapsulates both the data storage and parallel computing requirements for large scale and high-resolution air quality simulations that frequently rely on output generated by other models that are dependent on chosen science configurations. Specifically, numerical models for simulating regional and global scale air quality events are developed with a core function to support a variety of science configuration options that are enabled at compile-time in addition to a suite of run-time options. Efforts needed to treat such complex system models in the software as a service (SaaS) paradigm (Zhang et al., 2019) have remained exploratory and not gained enough traction, as correctly applying such models to specific situations demands a level of user control that goes beyond what is considered “power user” and involves administrative skills and in-depth HPC knowledge. This makes model deployment extremely difficult to achieve through a web-based interface. While end users have the option to use images with precompiled standardized versions of air quality models through CSP marketplace offerings at an hourly cost, these commercial products are designed for specific implementations, and their associated base science options.

As an example, Zhuang et al., (2020) demonstrated the scalability of GEOS-Chem to thousands of cores using the AWS ParallelCluster to achieve similar computational and cost efficiencies of local HPC clusters. They provided an easy-to-follow research workflow in an HPC cluster environment on the cloud. We extended this work by running the CMAQ model on Amazon Web Services (AWS) ParallelCluster and Microsoft Azure CycleCloud and using the HPC Cluster high-level frameworks or IaaC provided by these two major cloud providers. We provide tutorials that give end-users the ability to reproducibly provision HPC clusters and software in a way that is optimized to run CMAQ on the cloud in a turn-key service.

Furthermore, the increase in availability of large datasets in the cloud through vehicles such as NOAA's Big Data Program/ NOAA Open Data Dissemination (Simonson et al., 2022; NOAA's Big Data Program, 2023), CMAS's Data Warehouse on AWS (CMAS's Data Warehouse on AWS, 2023), and GEOS-Chem registry of open data (GEOS-Chem registry of open data, 2023) is another incentive to develop cloud solutions for air quality models that provide more leverage to the end-user. Such initiatives are critical for the mission and growth of cloud modeling and CSPs have acknowledged and addressed the emerging need of data democratization by waiving fees or providing free credits to facilitate access by scientists and average non-technical users of information systems. Tools such as AWS ParallelCluster and Azure CycleCloud are services that extend the power of IaaS by mimicking on-site HPC setups and provide an even more dynamically scalable environment that enables CMAQ modelers to step beyond the limits of single virtual machines (VMs), using the Simple Linux Utility Resource Management (Slurm) (Yoo et al, 2003) batch scheduler in a way that enables auto-scaling of the compute nodes, simplifying the cluster deployment and management. It is important to emphasize, that ParallelCluster and Azure CycleCloud extend the capability from simply being able to run on a VM hosted in the cloud to a turn-key batch scheduling environment that is dedicated to the end-user.

The Community Multiscale Air Quality (CMAQ) Model (Byun and Schere, 2006; Foley et al., 2010; Appel et al., 2017, 2021) is an open-source modeling system that consists of a family of programs for conducting air quality simulations and being actively developed. The Community Modeling and Analyses System (CMAS) Center facilitates community model development by hosting, developing and distributing software such as the CMAQ model, hosting the CMAS Center User Forum to facilitate exchange of information related to code and datasets and troubleshooting, and providing outreach and support through new user training, annual conferences and workshops on specific topics. In many cases, one or more factors are increasing resource requirements for CMAQ simulations, including the addition of more complex algorithms to CMAQ, simulations of longer time periods or larger domains, and modeling grids with finer resolutions. For institutions that use traditional HPC centers, despite the evolution of job managers, resources frequently come with allocation time limits and long queue times. Even if groups can afford to acquire and maintain appropriate computing capacity, such an approach may not be cost-effective, especially if the capacity is not fully utilized. By leveraging cloud infrastructure, CMAQ users can pay monthly on-demand fees to perform model simulations on clusters managed by commercial providers, without having to pay large up-front costs to purchase computer clusters or hire staff to maintain them. This can be extremely useful in enhancing computationally demanding research and air quality forecasting at an international scale, in many cases offering unprecedented expansion of such capabilities for developing nations. Another advantage of this approach is timely access to cutting edge processors, that otherwise would require disproportionate wait time, resources, and effort to obtain. Similarly, scalability can be expanded in real-time and with minimal effort.

The purpose of this study is to demonstrate the efforts required to bring the CMAQ model version 5.3.3 (U.S. EPA, 2023) to the cloud and perform air quality simulations efficiently and affordably, leveraging existing and publicly available datasets. In the following sections, we describe several key aspects of this work:

- Develop benchmark test suites that can address and replicate the needs of a typical CMAQ user
- Streamline the CMAQ installation process in Amazon's AWS and Microsoft's Azure
- Demonstrate running CMAQ on the cloud and estimate associated costs, making suggestions on different options available to the modeling community
- Perform benchmark tests with different HPC clusters and their underlying VMs, networking, and storage options while keeping track of the performance and associated costs
- Make recommendations that would help reduce CMAQ simulation times specific to the cloud platform.
- Provide instructions for obtaining and using input datasets from the CMAS Data Warehouse under the AWS Open Data Program which waives data egress costs

The methodologies used in this study are available as hand-on tutorials, with details for a variety of HPC systems on different CSPs, guides, and recommendations for specific user needs (see the links under *HPC Cluster Deployment Options*).

## 2. CMAQ Workflow and Cloud Benchmark Suite

Air quality modeling systems such as CMAQ rely heavily on the parameterization and simulation output from numerical weather prediction (NWP) systems in an offline coupling manner facilitated by pre-processing tools. Initial and boundary conditions for regional-to-urban scale simulations can be defined by the user to be either static or the result of nested downscaling from a coarser domain model application (i.e., Hemispheric CMAQ). A common CMAQ workflow involves:
1. developing meteorological fields with the Weather Research and Forecasting (WRF) Model (Skamarock et al., 2021),
2. processing WRF output using the Meteorology-Chemistry Interface Processor (MCIP) (Otte et al, 2010),
3. developing emissions inputs using the Sparse Matrix Operator KErnel (SMOKE) modeling system (Houyoux et al, 2000),
4. developing other inputs such as initial and boundary conditions using preprocessors,
5. performing air quality simulations using the complete set of inputs,
6. assessing the successful completion of the simulation and verifying the model output, and
7. analysis of the results to address the purpose of the simulation (e.g., regulatory or research issues).

Cloud storage enables reproducible workflows by having both model and datasets publicly available and directly accessed by the run scripts.

Traditionally, every CMAQ release is distributed with a lightweight test case that includes all inputs necessary for the user to confirm a successful installation and completion of a multi-day simulation. Similarly, a newly standardized test case, referred to as the cloud benchmark suite (CBS), was developed to evaluate CMAQ's performance on cloud HPC environments. Benchmark suite simulations were designed considering different user needs and data availability to construct a well-established bundle of inputs and outputs that can be further scaled and customized to meet specific scalable requirements. The hardware configuration necessary to run CMAQ depends on the domain size, grid resolution, complexity of physics and chemistry parameterization, number of variables and layers saved to the concentration and diagnostic files, and simulation duration. Since typical input and output data sets for CMAQ include three-dimensional descriptions of the dynamical and chemical state of the simulated atmosphere, these datasets could require several gigabytes of disk storage per simulation day. Given these considerations, a two-day CBS for the contiguous United States (CONUS) was constructed with the aim to be representative of a commonly used domain over a time frame that can be used to fully test the CMAQ system.  Typical requirements for a CONUS 12-km x12-km horizontal grid resolution are provided in Table 1 below, while Figure 1 shows a map with the domain's coverage. The storage space requirements are defined based on the need to perform multiple sets of identical runs while changing the number of CPUs used to run CMAQ in single-node workflows or parallel HPC implementations using OpenMPI (Gabriel, et al., 2004). Additional benchmark simulations were performed using different MPI configurations to change the number of available processors in order to evaluate scalability. All simulations used a modified version of the CMAQ model version 5.3.3 with the CB6 chemical mechanism and aerosol module 7 (cb6r3_ae7_aq). Further details are provided later in this paper. Datasets are typically created in the NetCDF data format (Unidata, 2023) that allows for sharing on the cloud following programming methods that leverage the power of Models-3/EDSS Input/Output Applications Programming Interface (The BAMS/EDSS/Models-3 I/O API: User Manual, Related Programs, and Examples, 2023).  Benchmark runs were performed with two output options: the first using a fully enabled CONC output option (37 variables, 36 layers), the second with a reduced number of variables and layers saved to the output concentration (CONC file) (12 variables, 1 layer). The scaling benchmarks used the reduced file option because the I/O API in its current version is not parallelized, and using the full output file may have negatively impacted the compute portion of scaling.

**Table 1: CMAQ model configuration and storage needs for the CONUS case benchmark suite.**

| | |
|---|---|
| CMAQ version | 5.3.3 with code modifications to fix cloud-specific bugs |
| OS | Linux, Processor 64-bit x86 (Ubuntu on AWS, Almalinux on Azure) |
| Memory | >1 GiB RAM per CPU core |

| | |
|---|---|
| Storage | Disk space requirement for the 2-day Benchmark Suite is 250 GB: 44 GB Input data and 170 GB Output data (output files included concentrations for all species) (CBS_full) or 18 GB Output data (for CONC file limited to 12 species, 1 layer) (CBS_limited) |
| Domain (ncols x nrows x nlays) | 396 x 246 x 35 |
| Horizontal domain resolution | 12-km x 12-km |
| Temporal resolution of output | Hourly |
| Temporal duration | 2 days |
| Chemical mechanism | cb6r3_ae7_aq |

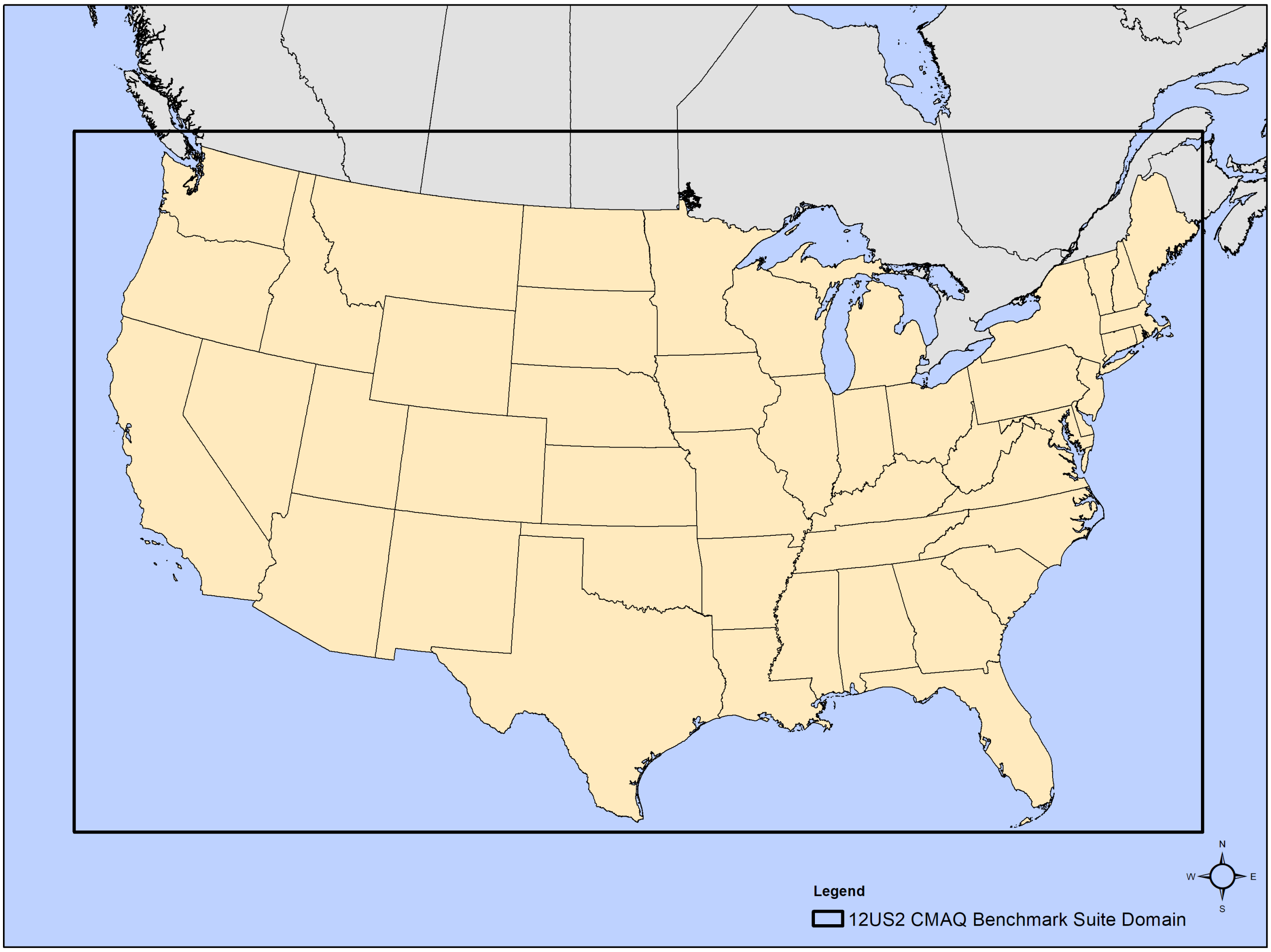

**Figure 1: Cloud benchmark suite modeling domain ("12US2"; 396 columns × 246 rows × 35 vertical layers) for the CONUS at a 12-km x 12-km horizontal grid spacing is shown as the bold rectangle.**

## 3 CMAQ Technology Stack

### 3.1 CMAQ Software Stack

*Model and prerequisite libraries:* Installing and setting up CMAQ on different CSPs with comparable Linux operating systems follows the general method depicted in the schematic of Figure 2. Step-by-step instructions to install the software stack using automated C-shell scripts are provided in the online tutorials. In addition, the tutorial covers the preparation of the benchmark

data and provides run scripts for launching CMAQ through the job manager. To facilitate an even better approach, publicly available snapshots of the /shared volume that contains the software stack are provided for each CMAQ model and hardware release. This allows new users to build clusters and quickly run CMAQ on HPC systems on the cloud. Additionally, it allows users to directly invoke existing precompiled libraries as modules, allowing for multiple applications and versions to be used and speed up model workflows and modifications (https://modules.readthedocs.io/en/latest/).
Depending on networking and storage options, users may need to add specific drivers and/or filesystem clients/layers to the list of installed modules. In parallel filesystem cases like Lustre, a client that is OS-specific needs to be present and linked to a storage account associated with the cluster to proceed for Azure Cycle Cloud. AWS offers a built-in Lustre implementation for most of their VMs including ParallelCluster. Azure VM images with embedded Lustre clients linked to a Lustre volume, currently in a public beta testing phase, were made available for our benchmark cases.

*Data Transfer Options.* AWS VMs have the AWS Command Line Interface (CLI) that is used to copy data from the S3 buckets available to the public through the AWS Open Data Program. For the case of Azure, users are provided instructions to install and use the AWS CLI and a *csh* script to copy the data from the CMAS Data Warehouse on AWS Open Data S3 bucket to the storage option being used. Data could also be copied from non-public S3 buckets to which the user has access privileges. An alternative is to link the S3 bucket to Lustre on AWS or create blob storage on Azure, and connect that blob storage to Lustre directly, to speed up access to input data. Azure users may want to use datasets from Microsoft's AI for Earth Data Sets (https://microsoft.github.io/AIforEarthDataSets/).

*HPC Cluster Deployment Options.* Step-by-step guidance for each CSP and the workflow used to run the benchmark has been documented and provided in the following tutorials (Azure: https://cyclecloud-cmaq.readthedocs.io/en/latest/ AWS: https://pcluster-cmaq.readthedocs.io/en/latest/). A verbose section was included in the run script structure to allow for recording architecture and OS specific parameters in the log files, including higher precision timing tracking of each model process. Recommendations for optimal MPI process placement using the Slurm Workload Manager with pinning on Azure HB series VMs were established for CycleCloud applications (https://techcommunity.microsoft.com/t5/azure-high-performance-computing/optimal-mpi-process-placement-for-azure-hb-series-vms/ba-p/2450663). Process placement was also used for ParallelCluster applications on AWS, optimized for the Hpc6a series. In the process outlined in Figure 2, we have also included code profiling tools (e.g., ARM MAP – https://www.linaroforge.com/) which allow for a better understanding of code performance and optimization opportunities for various applications/problem sizes. In Figures 3 and 4, we present overview schematics of the single VM and cluster configuration in each CSP. With respect to storage options, we chose the naming convention ***/local*** to refer to running CMAQ and saving the output on a local SSD, ***/lustre*** for performing the simulations on the Lustre parallel file system on AWS and Azure, ***/shared*** for using Elastic Block Store (EBS) on AWS and built-in Network File System (NFS) volume with default configuration on Azure, and ***/data*** for Azure's external NFS share

option (for more information see: https://learn.microsoft.com/en-us/azure/storage/common/nfs-comparison and https://docs.aws.amazon.com/parallelcluster/latest/ug/SharedStorage-v3.html). In general, storage implementations are CSP-specific and have different performance characteristics and fine-tuning options.

*HPC Cluster Monitoring Options.* The AWS Cloudwatch (https://aws.amazon.com/cloudwatch) webpage interface was used to monitor and compare the throughput of the I/O on the EBS and Lustre file systems using full output 37 variables, and all layers in the CONC file. The Azure Monitor Metrics (https://learn.microsoft.com/en-us/azure/azure-monitor/essentials/data-platform-metrics) webpage interface was used to compare the latency and throughput of the I/O on the shared and Lustre file systems using the cloud benchmark suite (CBS_full).

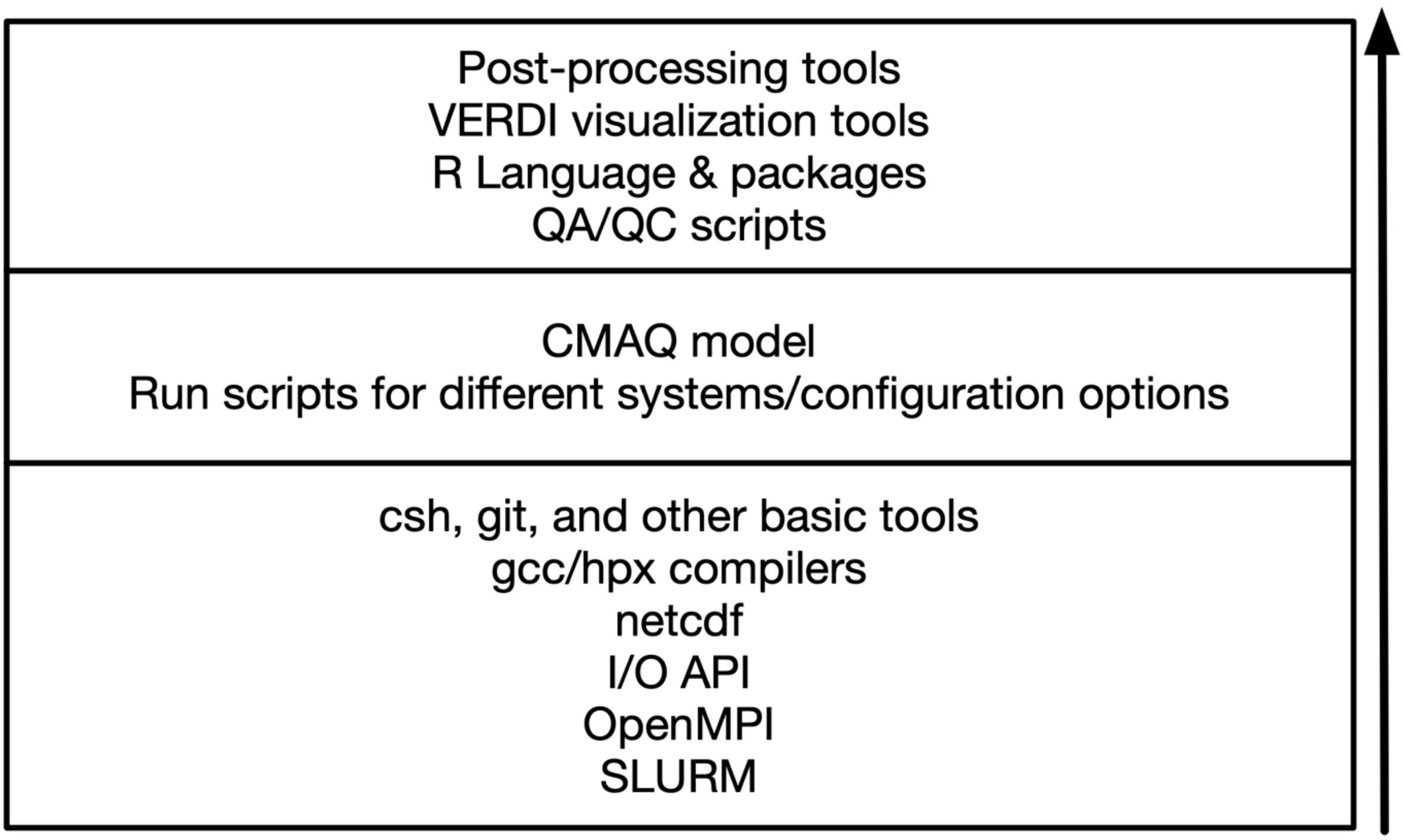

**Figure 2: Multi-step approach in installing CMAQ prerequisites, model code, post-processing, and other visualization/evaluation tools of the software stack.**

### 3.2 CSP Computation Options

The first step to begin using the cloud is to engage a cloud service provider (CSP) and create an account. This is the user's responsibility and CSPs have direct dedicated support to address specific user needs. Cloud-based CMAQ setups were

developed and are currently available on two CSPs: Amazon (AWS) and Microsoft (Azure). Typical CMAQ modeling workflows on the cloud can be divided into two general approaches: provisioning a single virtual machine and provisioning a dynamic multi-node cluster system. As multiprocessing architectures have evolved, many vendors are offering single VMs with more than 100 CPU cores, making them ideal for flexibly allocating and managing resources for computational simulations while limiting the effort required to compile and maintain the code/scripts. Clusters can be created in a multi-node framework following a similar approach, once access and availability of the total amount of resources is granted by the CSP.

After a thorough initial testing of the model code with a wide spectrum of hardware options offered by cloud vendors for HPC applications, we established the best performing architecture configurations described in Table 2 as the hardware stack test bed for final benchmarking under this study. Amazon's Hpc6a instances are powered by two 48-core 3rd generation AMD EPYC 7003 series processors built on 7nm process nodes for increased efficiency, 384 GiB of memory, 256 MB of L3 cache, with a total of 96 cores. AWS Nitro System offloads the hypervisor to a dedicated card that also provides high speed networking and high-speed Elastic Block Store (EBS) services, allowing all 96 cores of the AMD chip to be used for simulations (AWS, 2023). Azure's HB120v3 server features two 64-core AMD EPYC 7V73X processors for a total of 128 physical cores, while each section contains 8 processor cores with uniform access to a 96 MB L3 cache/section. The Azure HB 120v3 was designed to reserve eight cores for the Hypervisor and provides the remaining 120 cores for the application. Modern processors such as AMD's EPYC series employ Non-Uniform Memory Access, a multiprocessing (multi-die) architecture in which each processor is attached to its own local memory (called a NUMA domain) but can also access memory attached to another processor. To maximize the performance for each AMD chip it is important to balance the amount of L3 cache and memory bandwidth per core at the job level. This means that the binding of a process or thread to a specific core, known as CPU pinning or processor affinity, will now have to include additional steps for NUMA topology optimization (Podzimek et al., 2015; Ghatrehsamani et al., 2020).

### 3.3 Networking Options

In the tutorials and code implementations, we employed CSP-specific advanced networking options that reflect the available hardware options, enabling the 100 Gbps Elastic Fabric Adapter (EFA) on AWS, and the 200 Gbps High Dynamic Range (HDR) InfiniBand on Azure to support the level of high-performance computing required by CMAQ.

### 3.4 Storage Options

For storage, the real-time allocation of bandwidth and input/output operations per second (IOPS) differs between cloud vendors and should be examined independently at the application level by the user. In the examples investigated in this study, the user has access to four CSP-specific types of storage:

1. The fastest built-in local storage using Nonvolatile Memory Express Solid-State Drives (NVMe SSDs) that is included with default single VM provisioning

2. Network file systems tied to the user/enterprise account accessible using the Network File System (NFS) for Azure and Elastic File System (EFS) for AWS, attached to the head node in a cluster environment or directly to a VM

3. Unique services such as AWS's Elastic Block Store (EBS) which are designed for as per-instance block for certain compute cloud frameworks such as single AWS elastic cloud (EC2) and Azure's NetApp Files (ANF)

4. Fully managed high performance file system such as Lustre developed for HPC cluster environments (also tested with single VMs). Lustre implementations offer improved performance and allow for multiple compute servers to connect to the Lustre host, where several servers are responsible for handling the transfer.

*Summary:* We explored all the above options to have a complete set of solutions for different model cases and user needs that can be formulated around the cloud benchmark suite. In the standard CMAQ implementation, input is read by all available cores while output is handled by only one of them. While the model performed as expected with single VMs, the code base had to be modified to correct issues with NFS-mounted storage in cluster environments that utilize more than ~180 cores. The code changes did not have an impact on the model results. If a parallel file system is present (i.e., Lustre, BeeGFS), users have the option to configure CMAQ with the parallel I/O algorithm (Parallel I/O implementation of CMAQ, 2023). Such implementations for CMAQ have been explored in previous versions of the model code base and performance was investigated in more I/O demanding, higher spatially resolved simulations (Wong et al., 2015), but need to be thoroughly tested on the cloud with current compilers/hardware and were not considered at this stage of model benchmarking. It is, however, important to note that CMAQ input and output file sizes are highly dependent on the domain size and output file configuration options that can be simulation-specific, and users are encouraged to perform further analysis for their unique modeling application needs. Table 3 summarizes the different storage options that were included in the final set of benchmarks. This list does not include certain storage solutions such as Azure NetApp Files (ANF), common internet file shares (CIFS), and the BeeGFS parallel file system, as these options were deemed either too expensive or created challenges when benchmarking CMAQ, e.g., CIFS does not allow for file links, ANF was more expensive for the CMAQ paradigm compared to other offerings from Microsoft, BeeGFS is not available as a service and needs additional server setup and tuning.

**Table 2: Overview of system configurations and technical capabilities for the two HPC systems that were used for benchmarking.**

| **HPC Test System Description** | | |
|---|---|---|
| **Cloud Service Provider** | Microsoft Azure | Amazon Web Services |
| **Service Name** | CycleCloud | ParallelCluster |

| | | |
|---|---|---|
| **VM Name** | Standard_HB120rs_v3 | hpc6a.48xlarge |
| **Processor** | AMD EPYC 7V73X | AMD EPYC 7R13 |
| **CPU cores available** | 120 | 96 |
| **CPU speed (MHz)** | 1846 | 2650 |
| **Memory (GiB)** | 461 | 384 |
| **L3 Cache memory (MB)** | 96 | 192 |
| **Network Bandwidth (Gb/s)** | 200 (Nvidia HDR InfiniBand) | 100 (Elastic Fabric Adapter -EFA) |

**Table 3: Overview of storage options for the two HPC systems that were used for benchmarking.**

| **Storage Options** | | |
|---|---|---|
| **Cloud Service Provider** | Microsoft Azure | Amazon Web Services |
| **Service Name** | CycleCloud | ParallelCluster |
| **Storage option 1 (/local)** | Local NVMe SSDs in raid0 (2 * 960 GB NVMe - block) | N/A |
| **Storage option 2 (*/shared*)** | Built-in NFS: P30 Tier, Provisioned IOPS: 5000, Provisioned Throughput: 200 MB/s/TiB | Elastic Block Storage (EBS) – General purpose volumes (gp3): Provisioned IOPS: 3000, Provisioned Throughput: MB/s/TiB |
| **Storage option 3 (*/data*)** | NFS File share: Max IO/s 4024, Burst IO/s: 10000, Throughput rate: 203 MB/s | N/A |

| | | |
|---|---|---|
| **Storage option 4 (*/lustre*)** | Lustre 150 – size 128 TB<br>Performance profile:<br>150 MB/s/TiB<br><br>Lustre 250 – size 128 TB<br>Performance profile:<br>250 MB/s/TiB | Lustre SCRATCH_2 option:<br>Size: 1 TB<br>Network Throughput: 200 (1300 Burst) MB/s/TiB<br>240MB/s<br>Disk Throughput: 200 MB/s/TiB (read), 100 MB/s/TiB (write) |

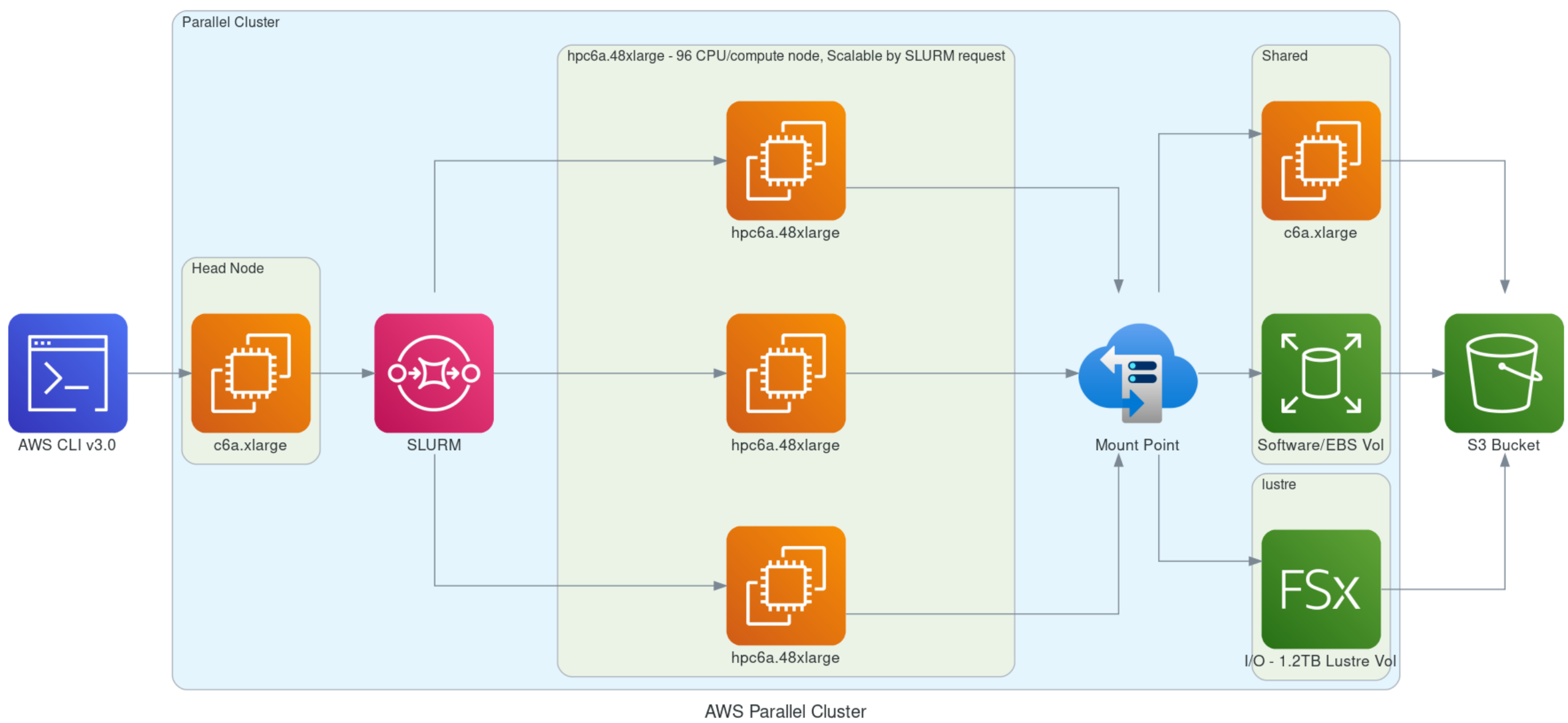

**Figure 3: Schematic demonstrating AWS's ParallelCluster-based framework utilizing different storage options.**

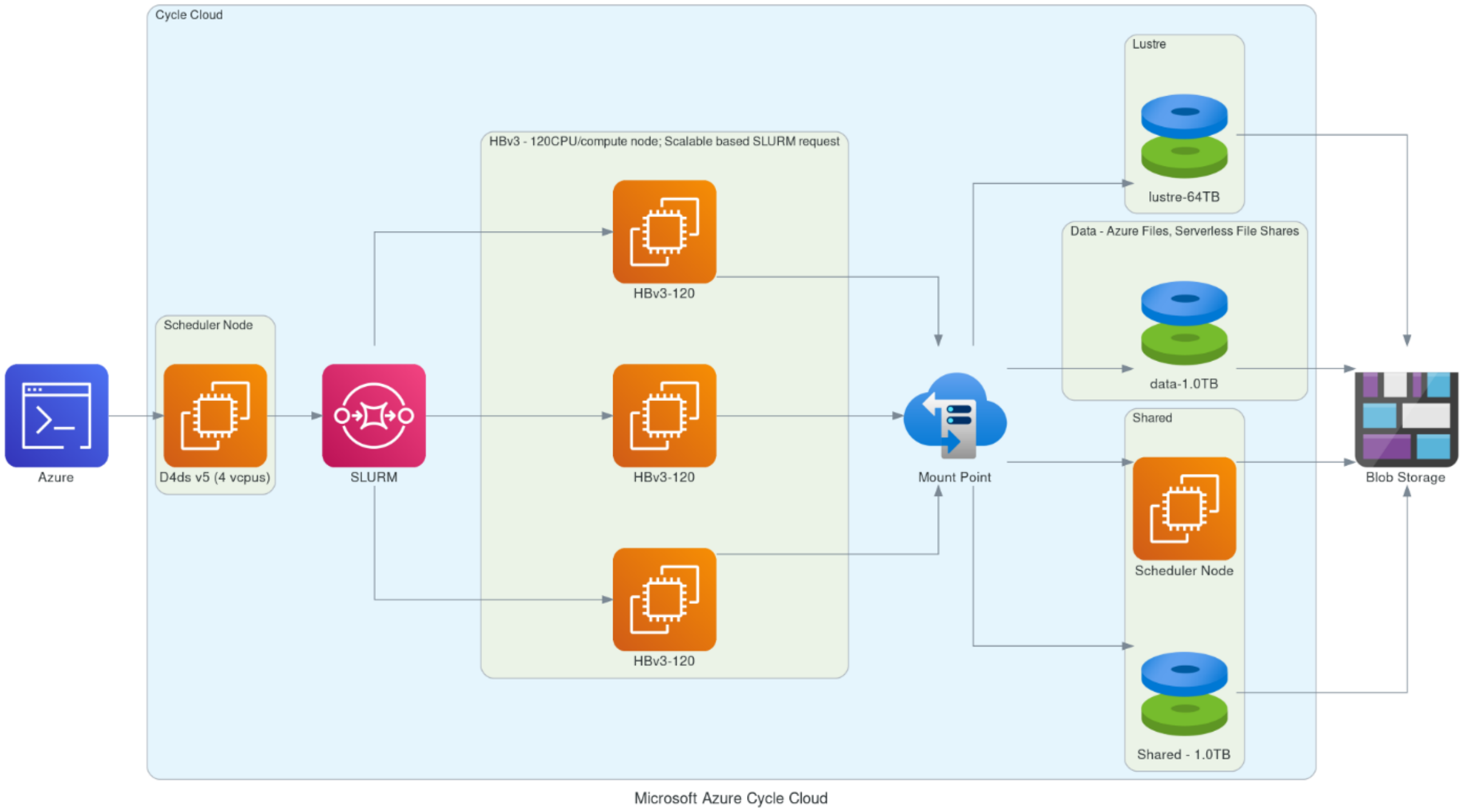

290 **Figure 4: Schematic demonstrating Microsoft's Azure CycleCloud-based framework utilizing different storage options.**

## 4 Results

### 4.1 Single-node VM Timing Analysis

The CMAQ simulation will write two types of logfiles, a main logfile and processor-specific logfiles for each core/process. Model performance was evaluated using the main logfiles that include timings for the major science modules at each timestep: Vertical Diffusion (VDIFF), COUPLE (Converts units and couples or decouples concentration values with the density and Jacobian for transport), Horizontal Advection (HADV), Vertical Advection (ZADV), Horizontal Diffusion (HDIFF), DECOUPLE, Photolysis (PHOT), Cloud Process (CLDPROC), Chemistry (CHEM), and Aerosol (AERO). At the end of each simulation hour, species concentrations are output along with the timings printed for the output process (Data Output). It should be noted, this output process timing does not fully capture the total I/O time including initializing and shutting down the model (i.e., close all files, deallocate arrays). This unaccounted time component is derived from the difference between the total wall time (elapsed real time) and the sum of the sub-processes and was labeled as OTHER in the plots.

Model scalability is the measure of a system's ability to increase (or decrease) performance (and cost) in response to changes in system processing power, in our case determined by the specific resources (cores, memory, storage, and network protocols and bandwidth), and relies on MPI implementation and integration with the job manager (Slurm). Results from the benchmark case simulations performed in a single-node EPYC VM of Microsoft's Azure are presented in Figures 5 and 6. Figure 5 demonstrates good performance and efficiency scaling with both local and NFS Solid State Drive (SSD) storage options and some degree of a leveling off observed above 96 cores. As expected, the fastest local NVMe solution performs better than the same system with different storage options. Since NVMe is included in the default configuration, it is also the cheapest solution for a testing phase and despite its fixed volume it is sufficient for the benchmark domain and simple user needs (i.e., benchmarking, code development, testing). For larger domains and simulation periods, the SSD over NFS is a preferred solution that allows for larger volumes of data to be attached. Figure 6 provides a cloud benchmark case performance comparison broken down by model process component for each storage solution within Azure. A difference in the VM core allocation for hypervisor and background system tasks results in a different core count available for compute between CSPs. For direct comparison with AWS, the system in both CSPs was optimized to utilize 96 out of the 120 available cores by employing process pinning, matching the region of best scalability observed at Figure 6. From this figure, it is evident that the Lustre filesystem competes very well with the local SSD solutions, followed by an additional performance difference for the NFS share (/data), and the proportionally slower but cheaper built-in NFS storage. On AWS, we could not provision a local SSD for the hpc6a.48xlarge single VM, so benchmark tests were limited to Lustre and EBS storage options. Results depicted in Figure 7 indicate that Lustre on AWS performed slightly slower compared to Azure, while the EBS option was a faster and more cost-effective solution for this CSP. However, users are advised to copy the input and output data in EBS (and local solution on Azure) as part of their workflow and additional time to complete the simulation should be accounted for.

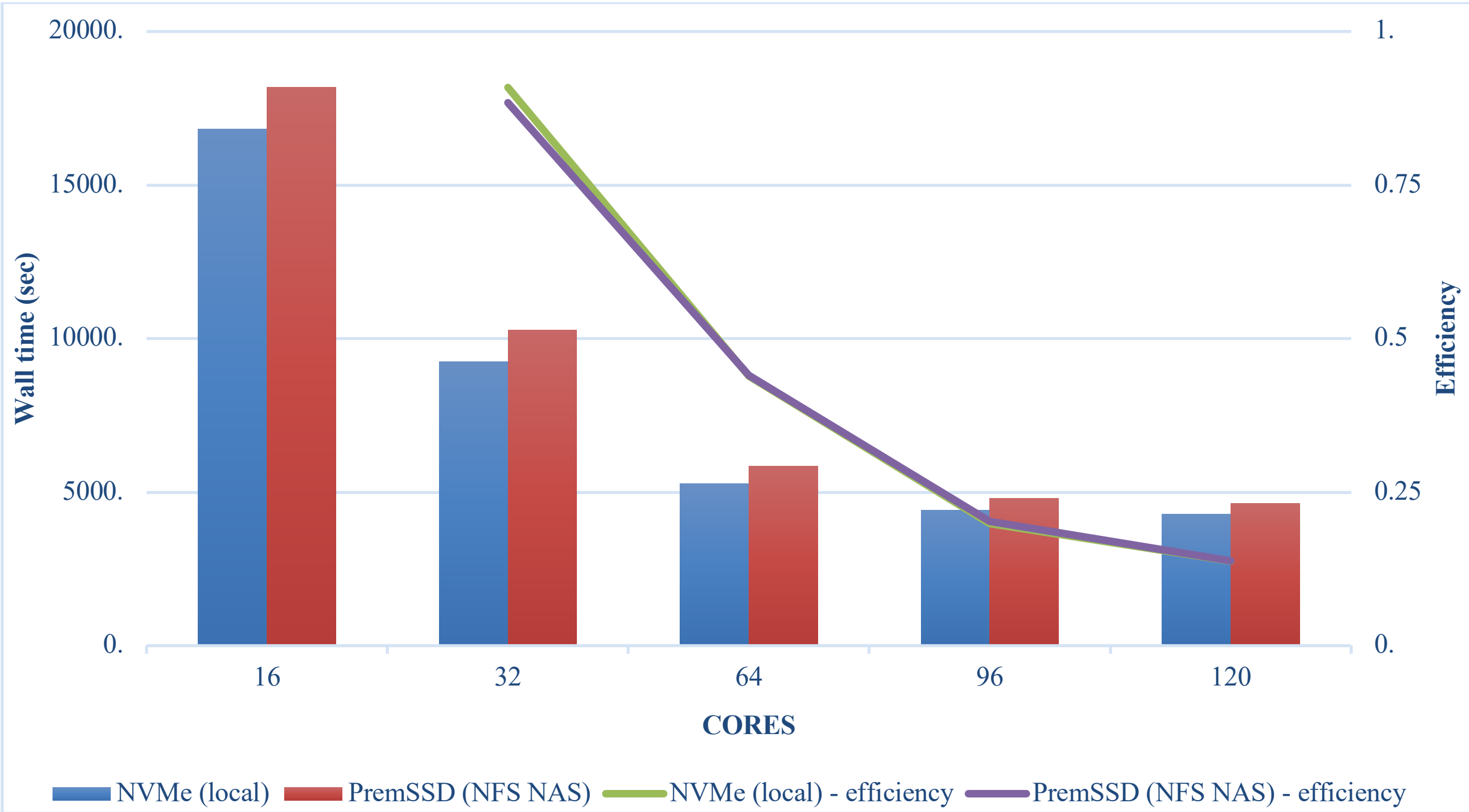

**Figure 5: Performance comparison of the cloud benchmark suite (CBS_limited) simulations on a single VM of Microsoft's Azure utilizing 16 – 120 cores with a fast local SSD NVMe and premium SSD through the NFS client.**

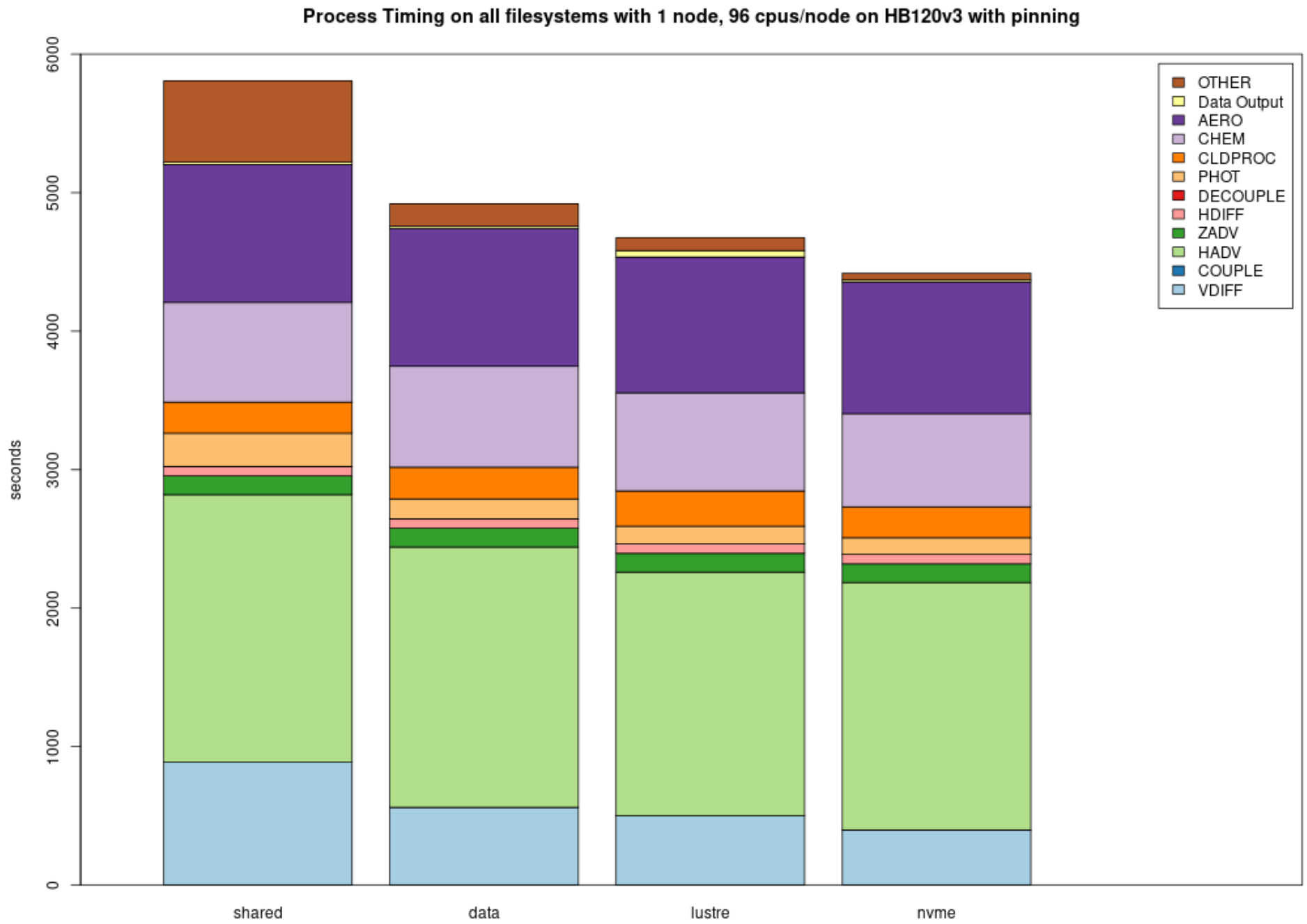

**Figure 6: Performance comparison per model component for the cloud benchmark suite (CBS_limited) on a single VM of Microsoft's Azure utilizing 96 cores with a fast local SSD NVMe, a Lustre file system, a NetApp (ANF) solution, and a premium SSD through the NFS client (from right to left).**

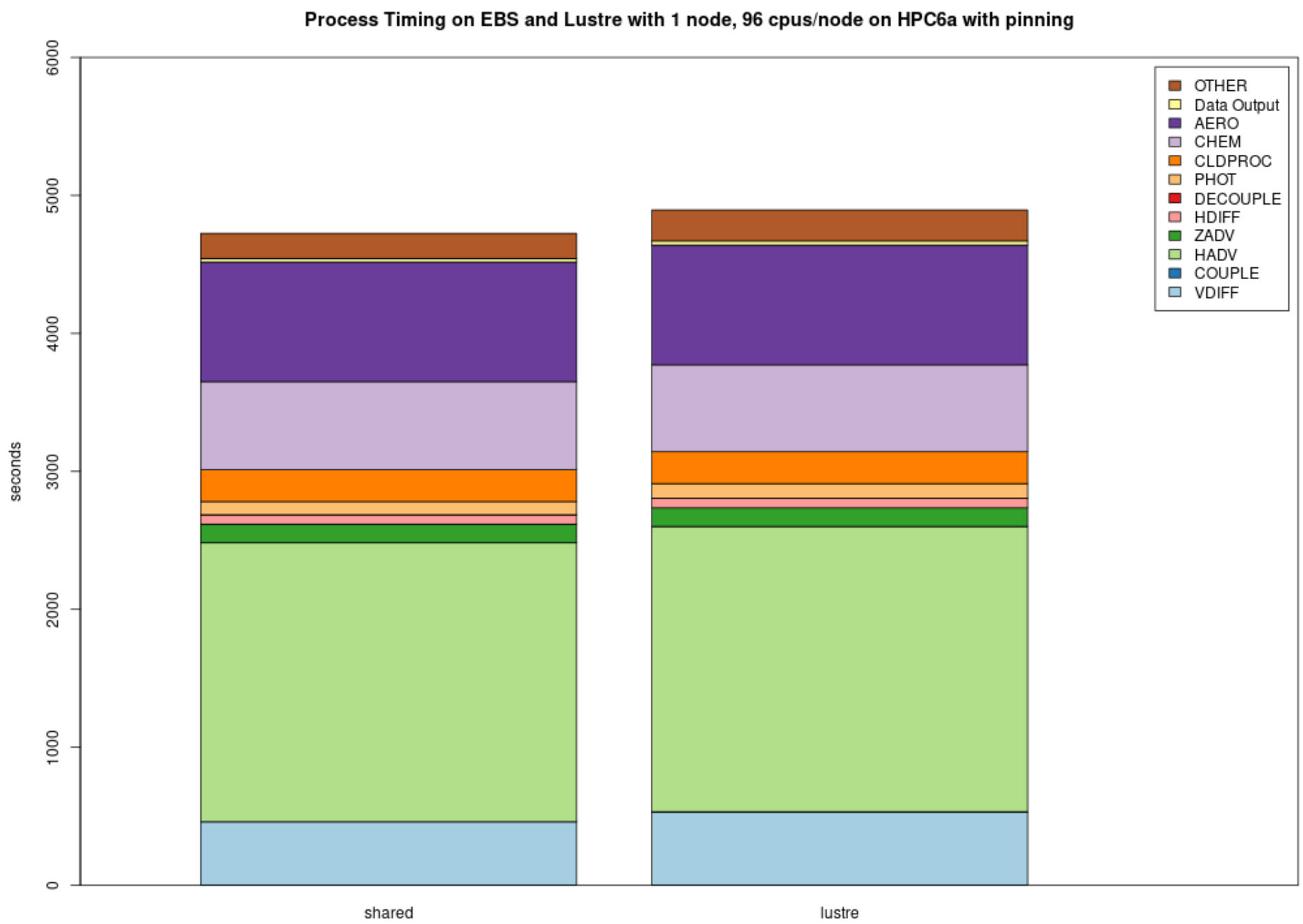

**Figure 7: Performance comparison per model component for the cloud benchmark suite (CBS_limited) on a single VM of AWS ParallelCluster utilizing 96 cores with the shared (EBS) and Lustre file system solutions.**

### 4.2 Optimization and Benchmarking on Multi-node Clusters

#### 4.2.1 Process Pinning for L3 cache Optimization in the EPYC Processor Architectures

As mentioned before, in a managed job environment, AMD EPYC processors offer an option called 'process pinning' that can improve performance through more effective use of the L3 memory cache at the job submission level. This is another configuration option that should be evaluated, especially since implementations vary between CSPs and may change over time. Figures 8 and 9 demonstrate the effect of process pinning on Azure and AWS, respectively. This option reduced simulation times on both systems using Lustre by approximately 20%. Figure 10 shows that this effect can vary depending on the filesystem as well, with EBS volume use pointing to more substantial performance gains in the AWS systems. Nevertheless, both Azure and AWS users should carefully consider such performance gains and further evaluate scaling under different pinning options for their domain and configuration.

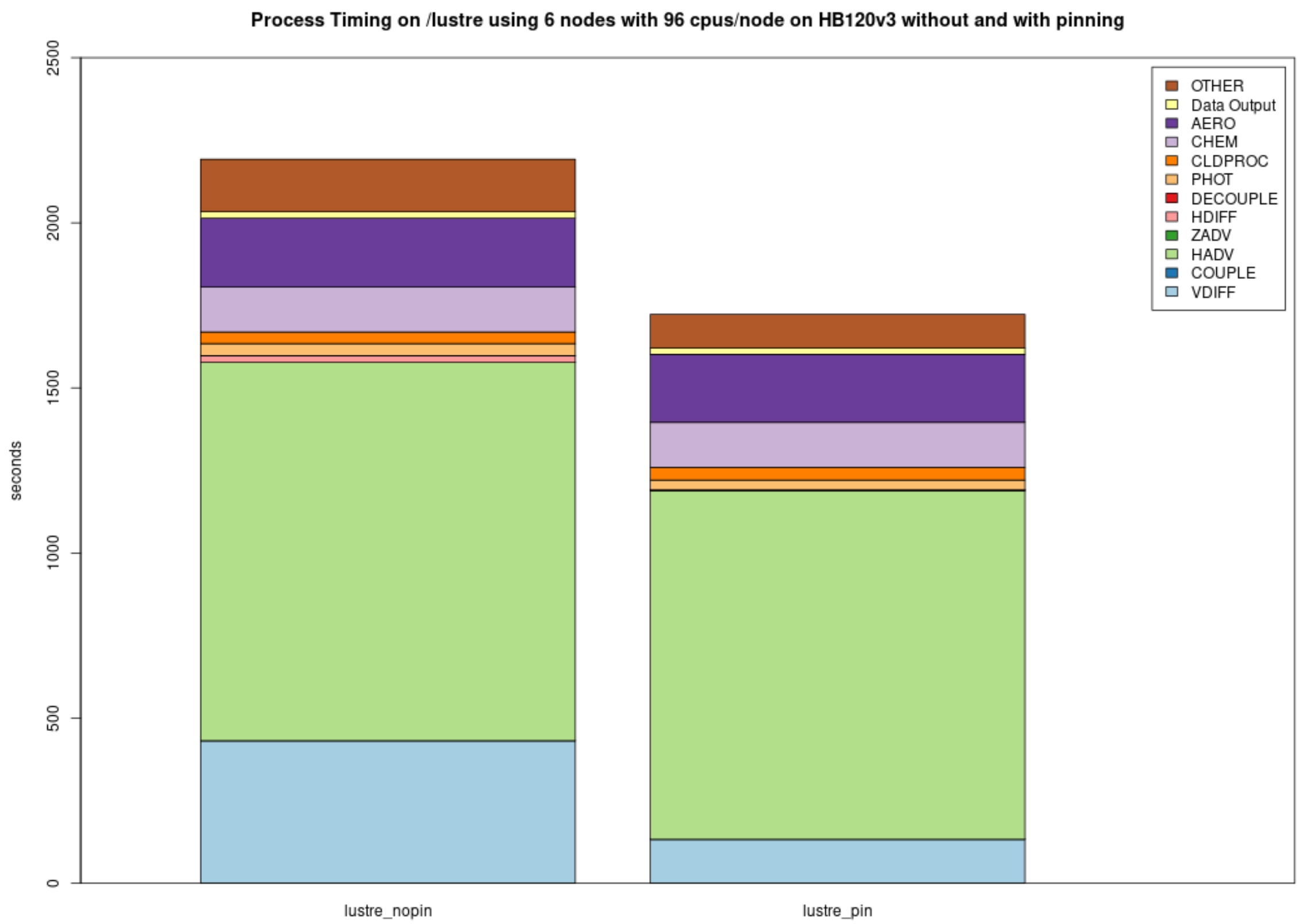

350 **Figure 8: Effect of process pinning on Azure CycleCloud (576 cores) on */lustre* for cloud benchmark suite (CBS_limited)**.

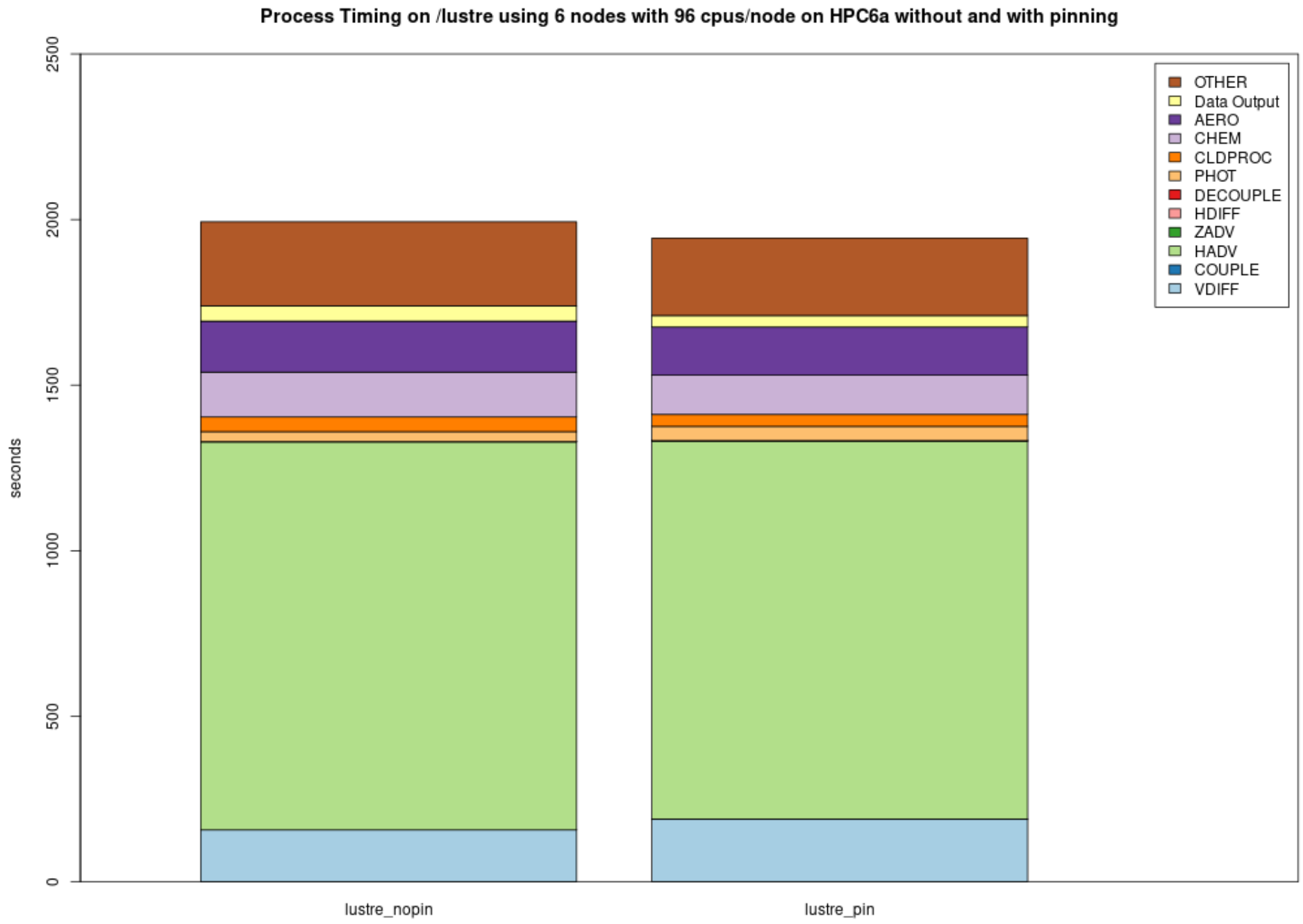

**Figure 9: Effect of process pinning on AWS ParallelCluster (576 cores) on *Lustre* for cloud benchmark suite**
355 **(CBS_limited).**

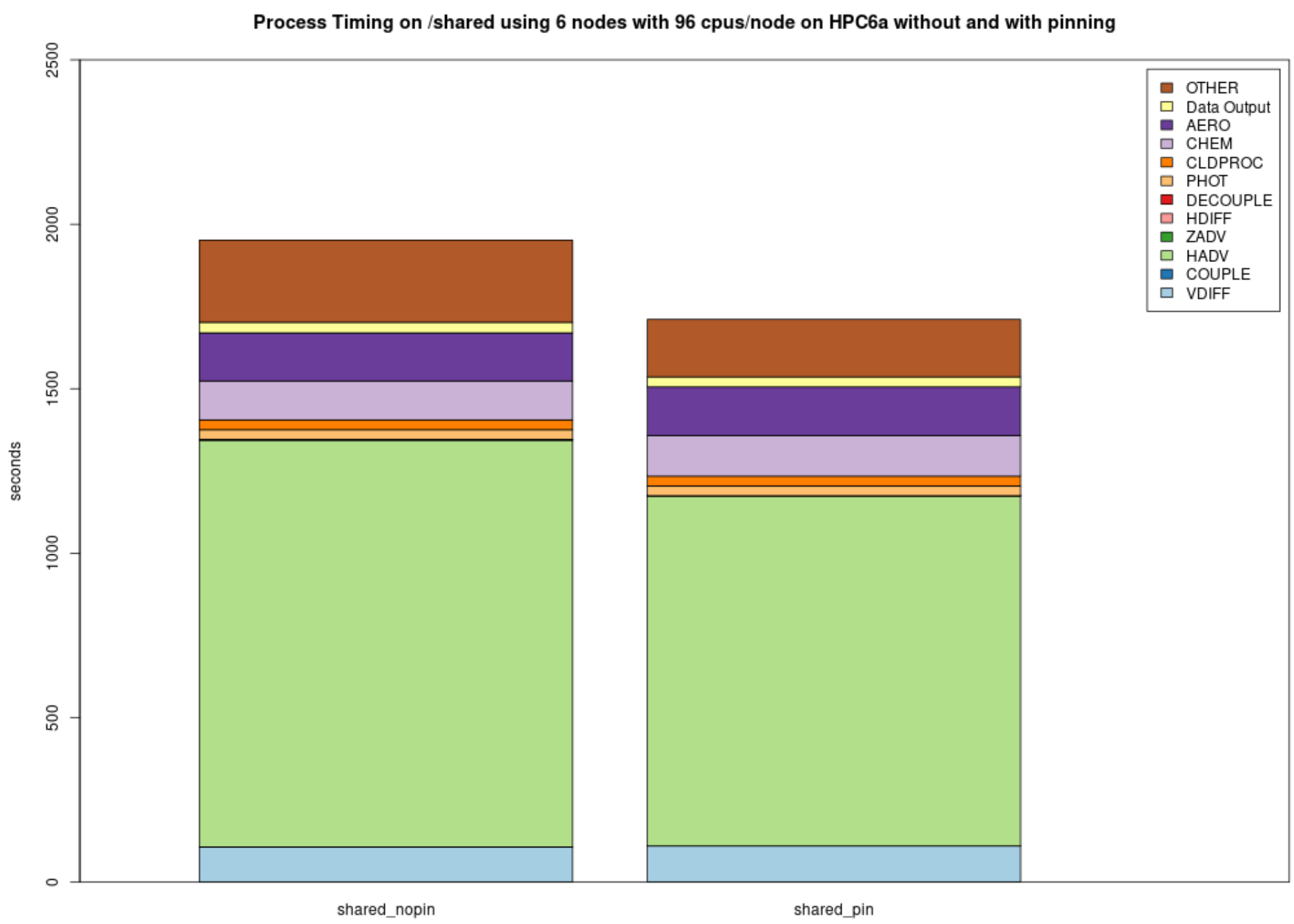

**Figure 10: Effect of process pinning on AWS ParallelCluster (576 cores) on *shared* (500 GB EBS volume) for cloud benchmark suite (CBS_limited).**

### 4.2.2 Results on Multi-node Clusters with Different Storage Options Saving 12 Variables to 1-Layer CONC File

Figures 11 and 12 demonstrate benchmark case results from simulations performed on Azure's CycleCloud clusters employing 1 – 6 nodes and two different Lustre implementations, a faster (250 MB/s/TiB) and a slower (150 MB/s/TiB) tier, both of size 100 TiB. Depending on the end-user cost and overall simulation needs, the slower solution can be a more cost-effective one, while the expensive option can be chosen when a faster turnaround time is necessary. Figures 13 and 14 show similar scaling results for the NFS share and the slower built-in NFS solution, respectively. In all cases, we observed diminishing performance gains when utilizing more than 2 nodes, with a plateau becoming apparent in the 3 – 6 node region. Figure 15 provides the performance breakdown for AWS's Lustre storage option, while Figure 16 provides the equivalent using the EBS (***/shared***) storage option. On AWS ParallelCluster, a scratch Lustre option (200 MB/s/TiB) was used. Lustre appears to be comparable in both CSP implementations, with minor differences that can be attributed to size of the filesystem that was provisioned (100 TiB on Azure Cycle Cloud, 1 TiB on AWS ParallelCluster), and the way the filesystem was parameterized (including stripe size) by different CSPs. EBS benchmarks were significantly faster, which makes it a potentially better alternative to Lustre for AWS instances.  On AWS ParallelCluster, the Lustre filesystem is connected to the CMAS Center Open Data S3 bucket, and

only the files that are used in the run script are copied from the S3 Bucket to Lustre. This strategy is used to identify resources as non-blocking (non-critical) and load these only when needed, referred to as lazy loading. For the EBS benchmark, an AWS CLI script is used to copy the data from the S3 bucket to the EBS volume. The time taken to copy the data using the AWS CLI is higher (~15 min) than the time it takes for the data to be read from the S3 Bucket by Lustre (~300 seconds) and these timings were not included since the data were pre-loaded to the filesystems for these benchmarks .

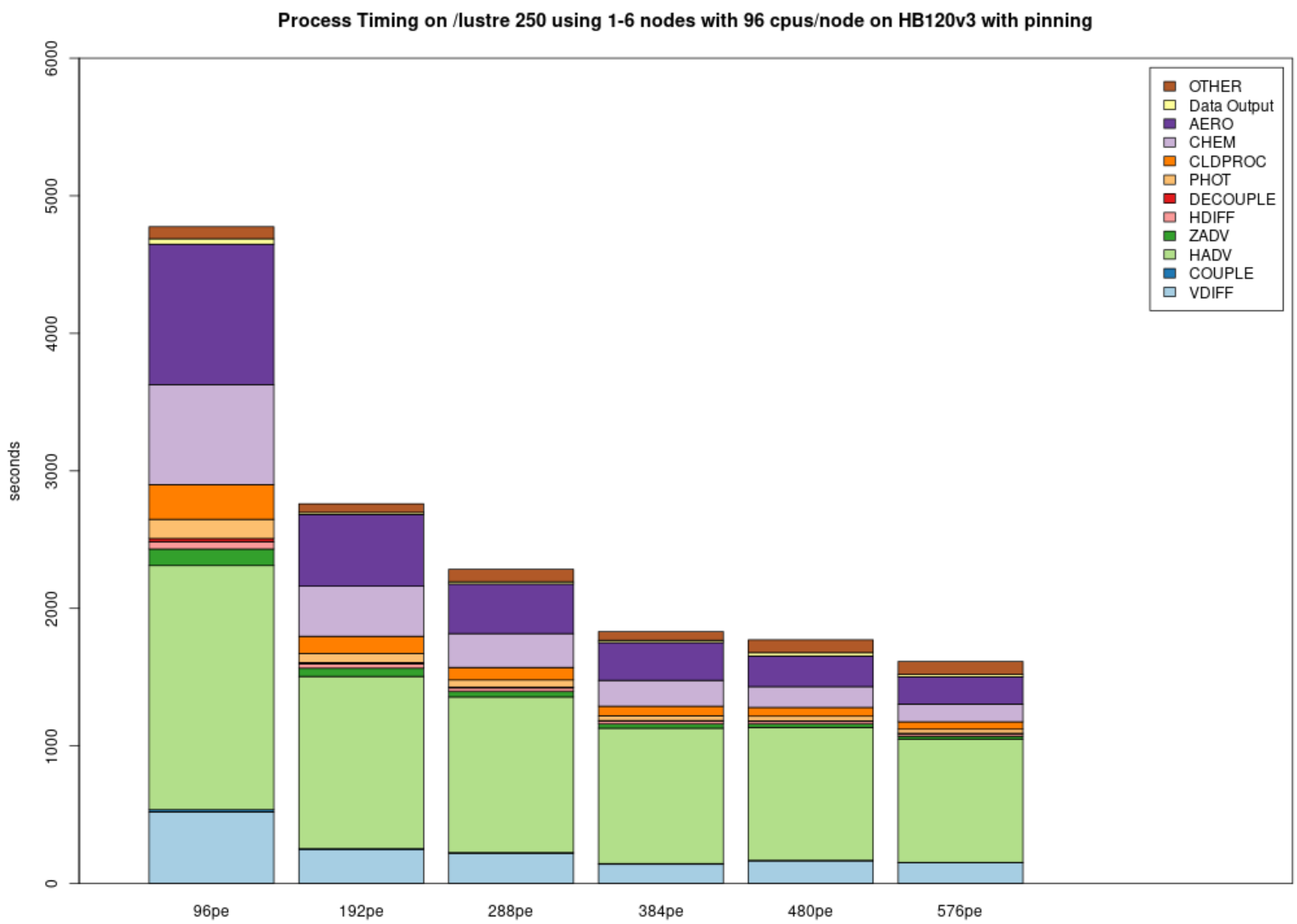

**Figure 11: Performance of the cloud benchmark suite (CBS_limited) on Microsoft Azure CycleCloud environments using a Lustre 250 filesystem for I/O and code on */shared*. Processing Element (PE) used in the x-axis label is equivalent to the number of cores.**

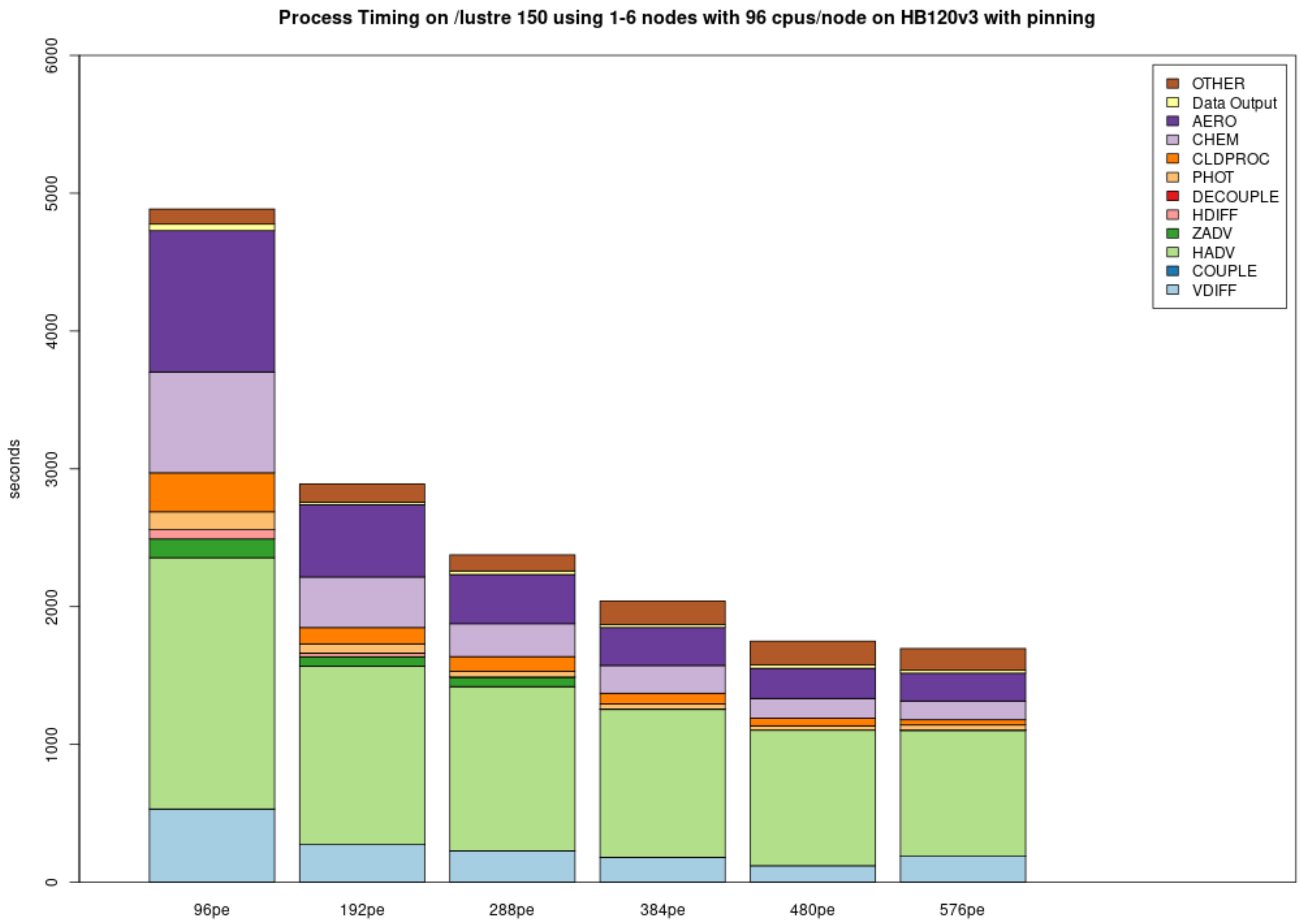

385 **Figure 12: Performance of the cloud benchmark suite (CBS_limited) on Microsoft Azure CycleCloud environments using a Lustre 150 filesystem for both I/O and code. Processing Element (PE) used in the x-axis label is equivalent to the number of cores.**

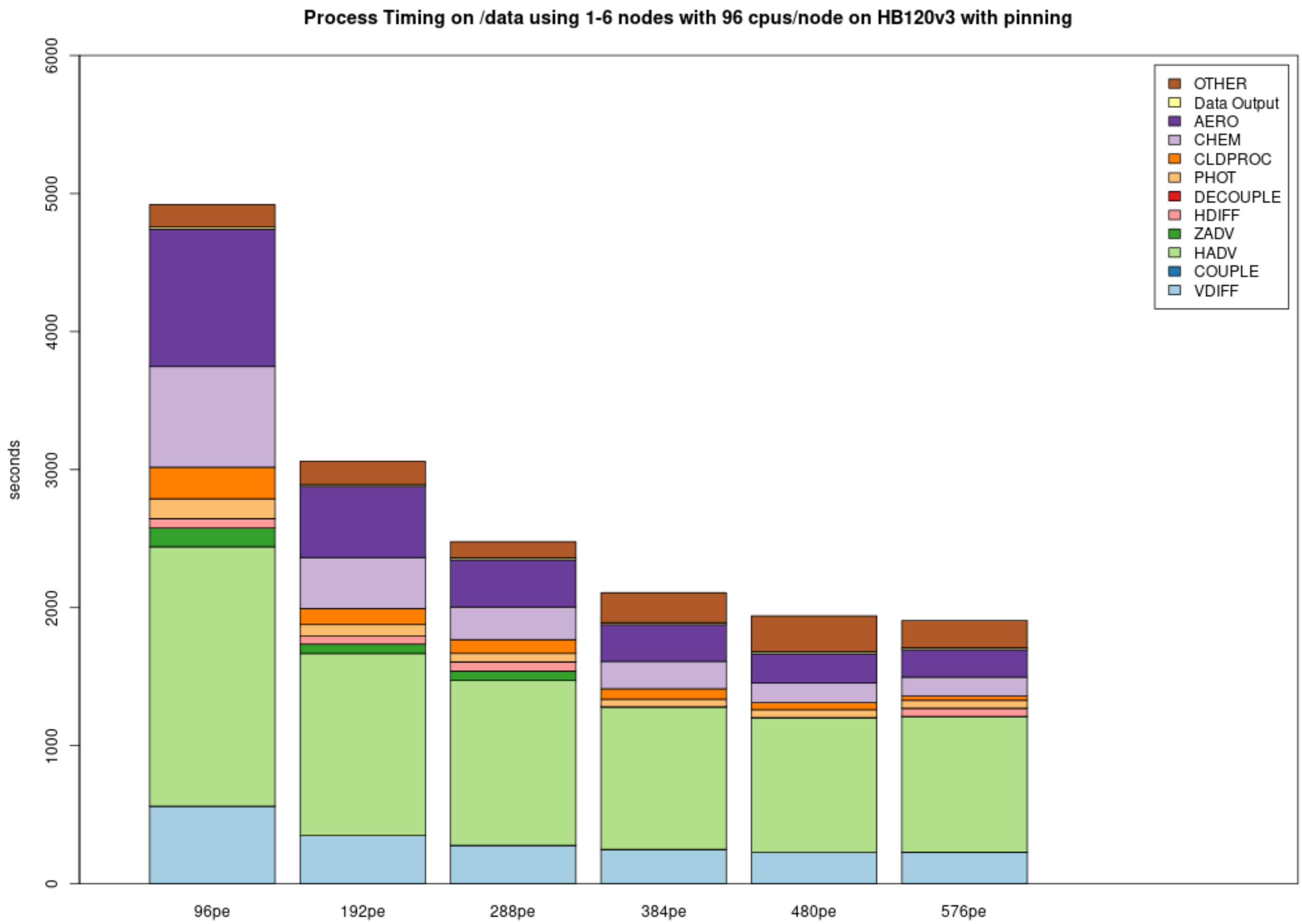

**Figure 13: Performance of the cloud benchmark suite (CBS_limited) on Microsoft Azure CycleCloud environments using the */data* filesystem for I/O and */shared* for the code. Processing Element (PE) used in the x-axis label is equivalent to the number of cores.**

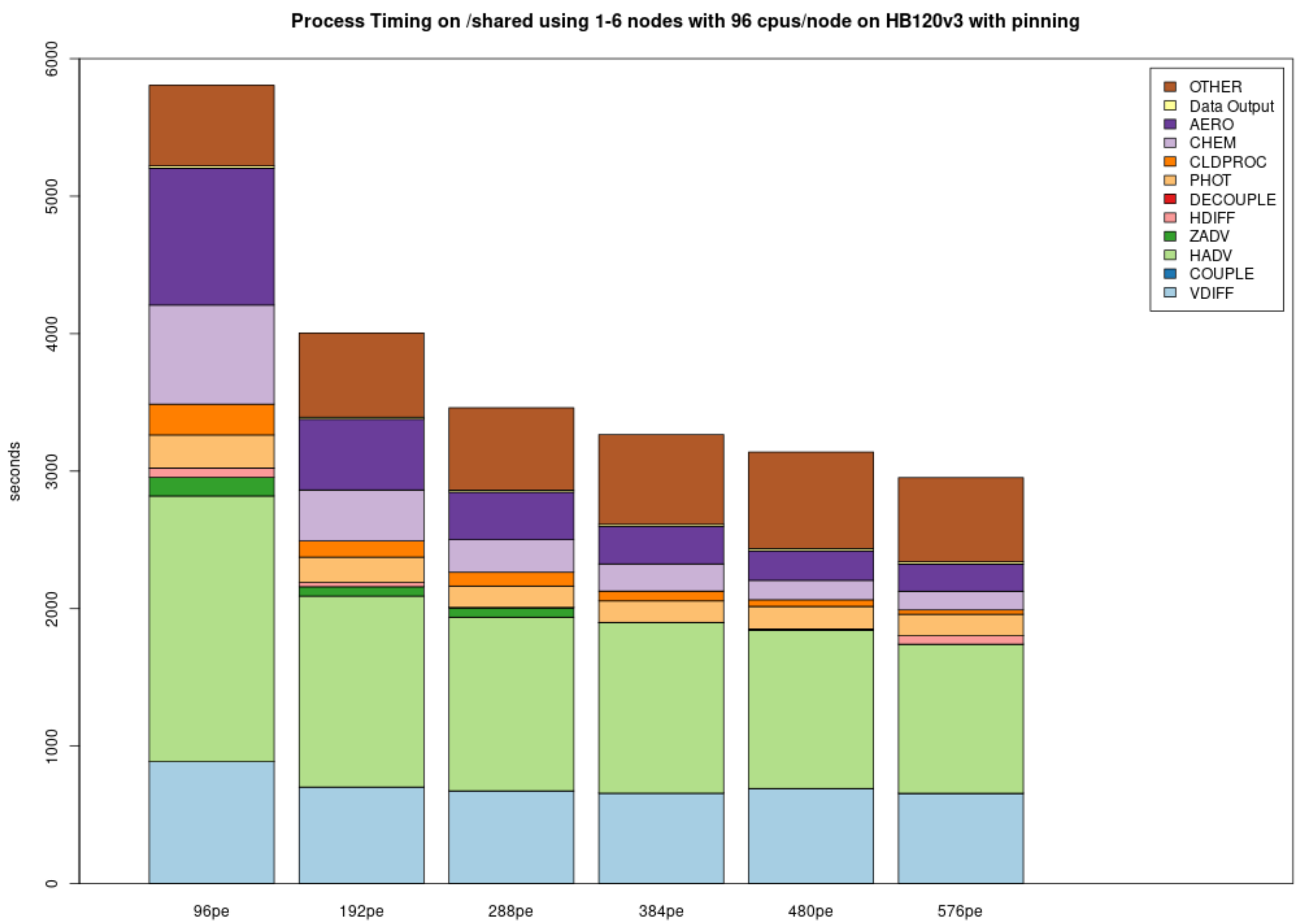

**Figure 14: Performance of the cloud benchmark suite (CBS_limited) on Microsoft Azure CycleCloud environments using a NFS built-in volume (*/shared*). Processing Element (PE) used in the x-axis label is equivalent to the number of cores.**

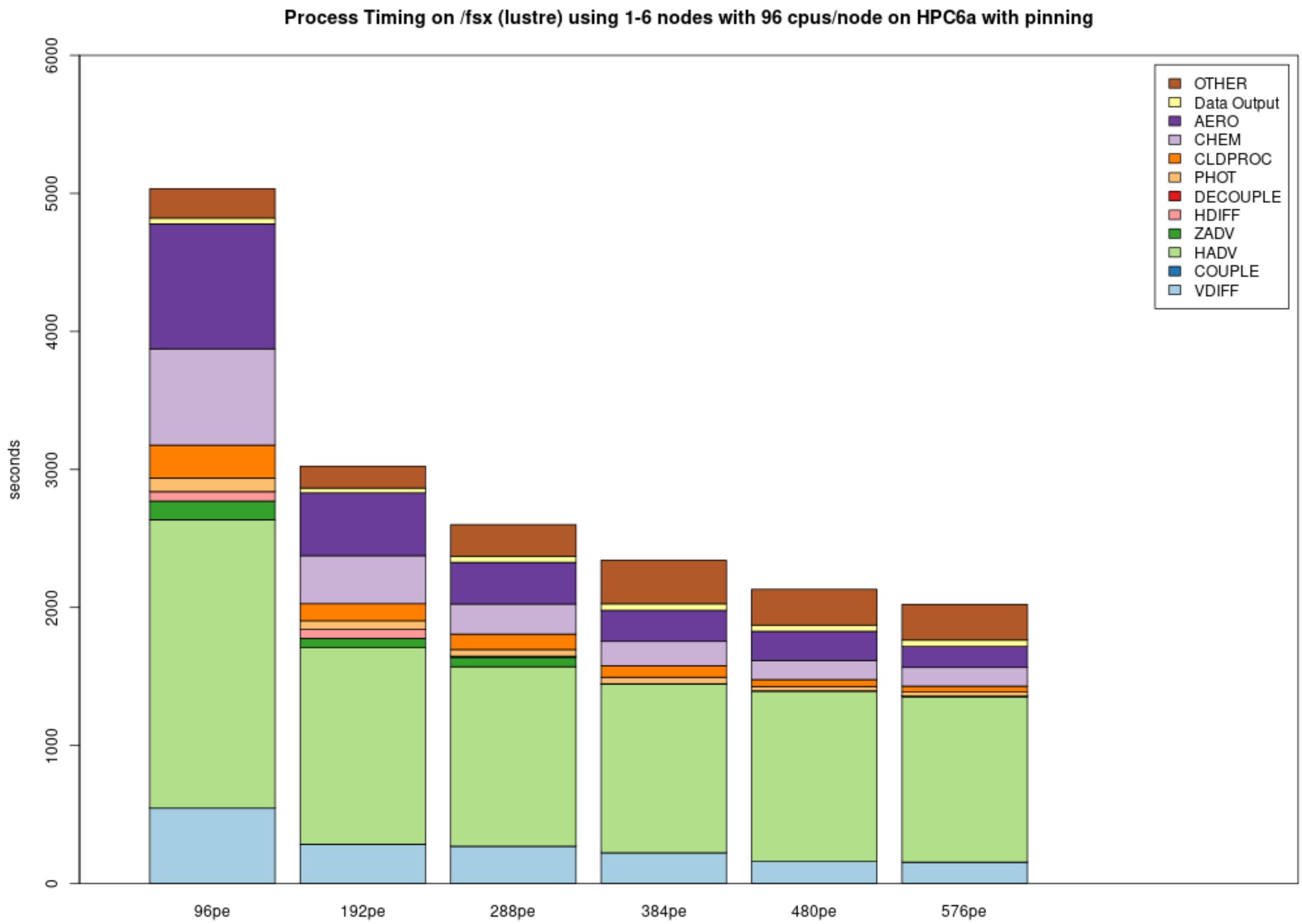

405

**Figure 15: Performance of the cloud benchmark suite (CBS_limited) on AWS ParallelCluster environments using a *lustre* filesystem. Processing Element (PE) used in the x-axis label is equivalent to the number of cores.**

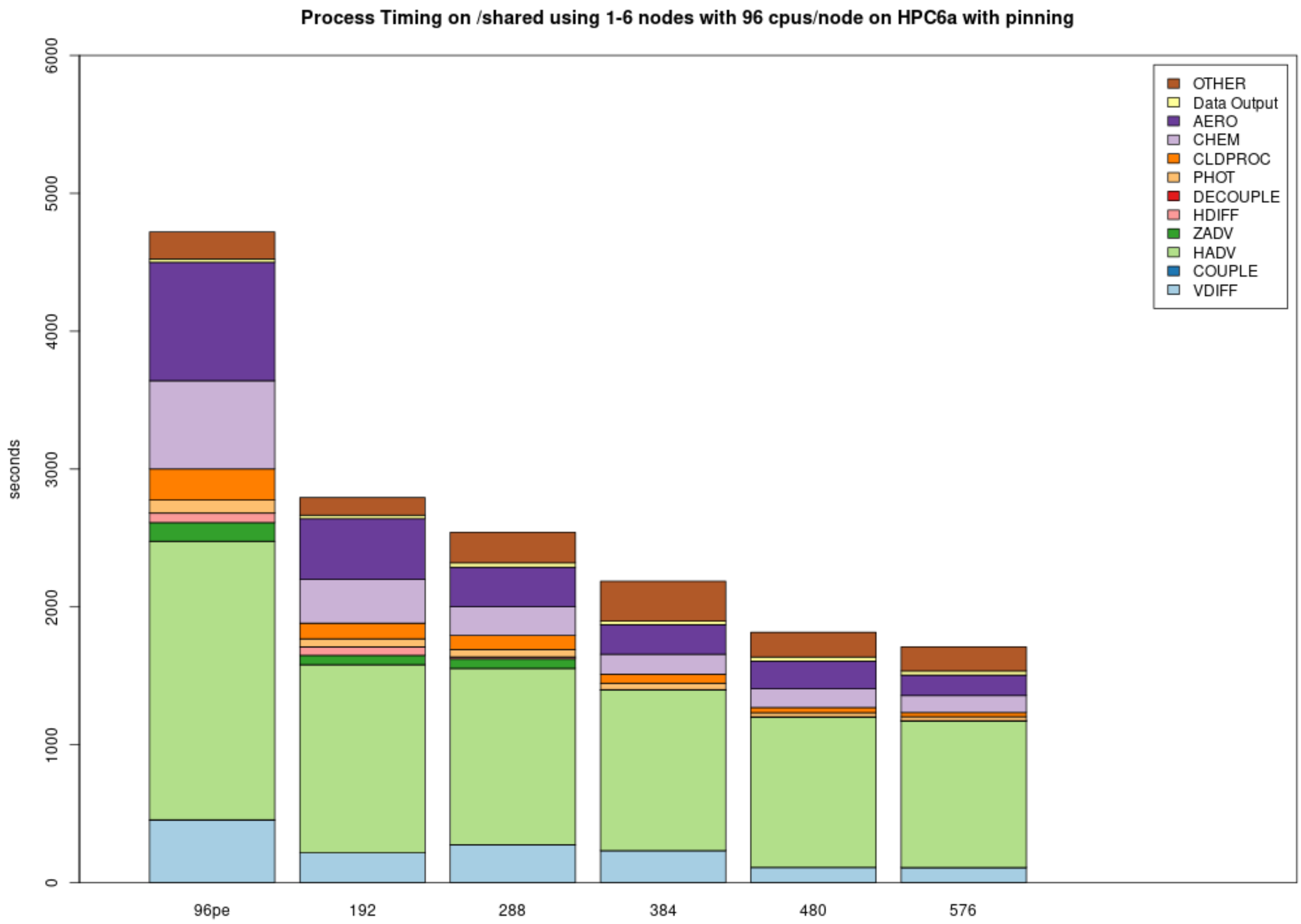

410

**Figure 16: Performance of the cloud benchmark suite (CBS_limited) on AWS ParallelCluster environments using a *****/shared***** EBS filesystem. Processing Element (PE) used in the x-axis label is equivalent to the number of cores.**

415 **4.2.3 Results on Multi-node Clusters with Different Storage Options saving 37 Variables to All-layer CONC File**

Figure 17 shows the performance of the EBS and the Lustre file systems using 96 PEs on AWS when the full number of output variables and layers are saved to the 3D CONC file (creating and saving the largest output file possible under the cloud benchmark case – CBS_full).

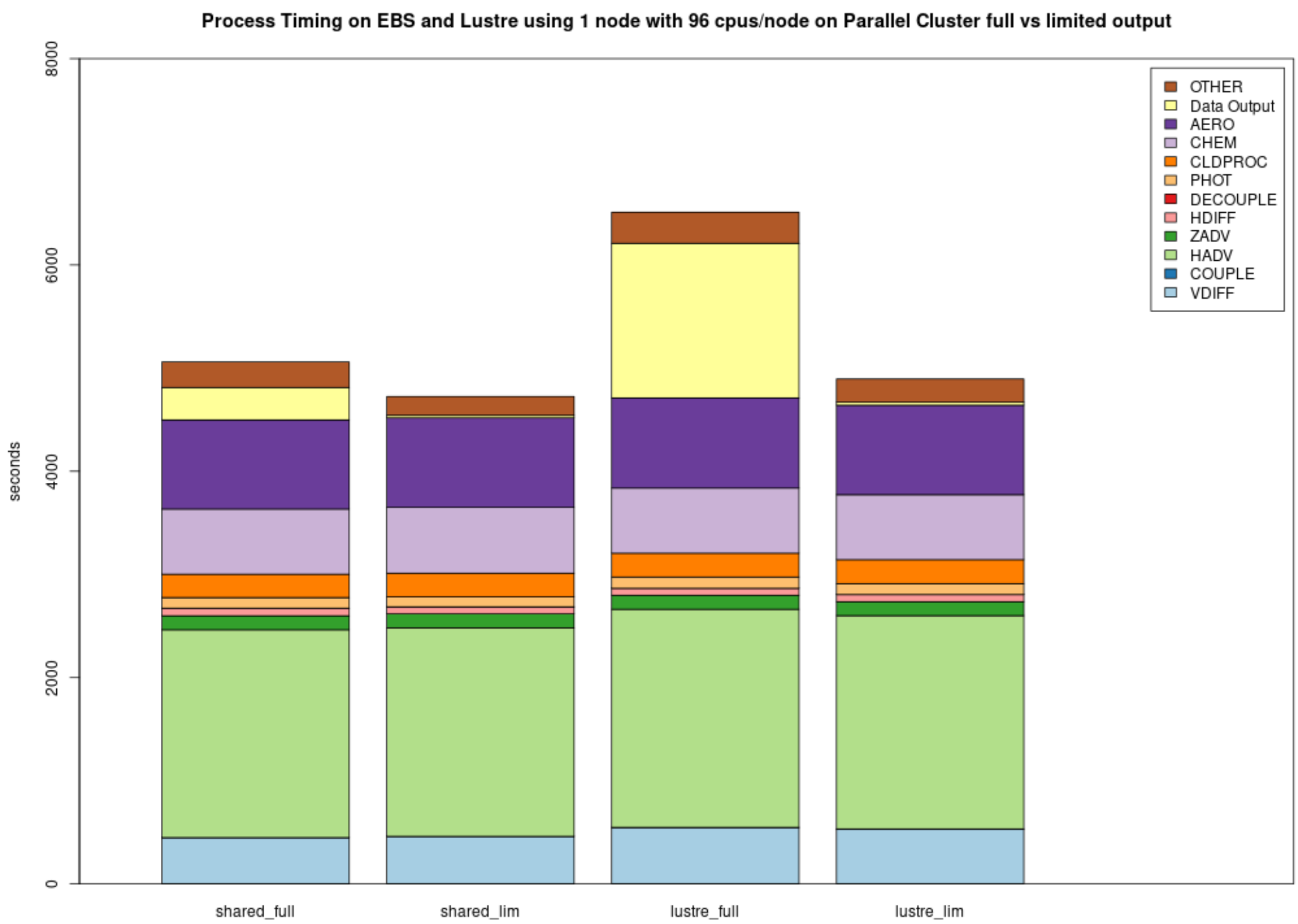

**Figure 17: Performance of the 2-day cloud benchmark suite on AWS ParallelCluster environments using full output (CBS_full) versus limited output on EBS (shared) and Lustre (CBS_limited).**

### 4.3 Amazon CloudWatch Throughput Monitoring for AWS ParallelCluster

Amazon CloudWatch (Amazon CloudWatch, 2024) and Azure Monitor™ (Azure Monitor, 2024) allow you to monitor the I/O throughput of a filesystem while running an application in real time. Amazon CloudWatch output shown in Figure 18 can be used to further investigate the disparities seen when writing additional model outputs (e.g., full layered 3-D instantaneous model concentrations) to different storage options. Figure 18 displays the results of using AWS Cloudwatch to monitor two benchmark runs using full variables and layers in the CONC file using ***/shared*** and ***/lustre***. While AWS Cloudwatch shows higher throughput on Lustre than shared, the benchmark performance is faster on shared than Lustre. This may be due to larger disk caches or faster latencies on the EBS volume. The Lustre performance may be improved by using a persistent volume

versus scratch volume that was used in this study. Figure 19 displays the read and write throughput, and client latency metrics from the Azure Monitor for the cloud benchmark suite using Azure Managed Lustre (250 MB/s).

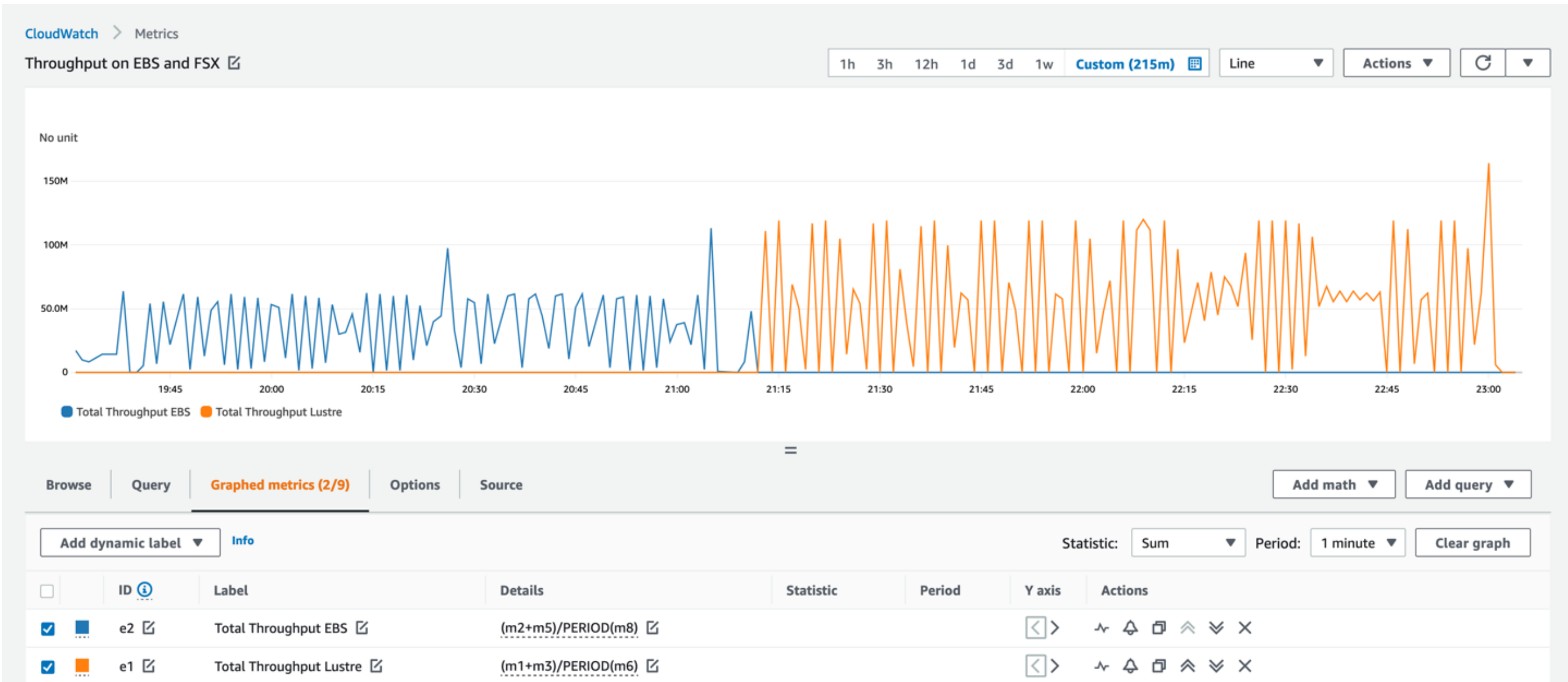

**Figure 18. Amazon™ CloudWatch™ throughput measurement of the first day of the cloud benchmark suite using 96 cores on ParallelCluster using different filesystems, EBS (blue) and Lustre (orange) for full output (all variables, all layers) in CONC file (CBS_full).**

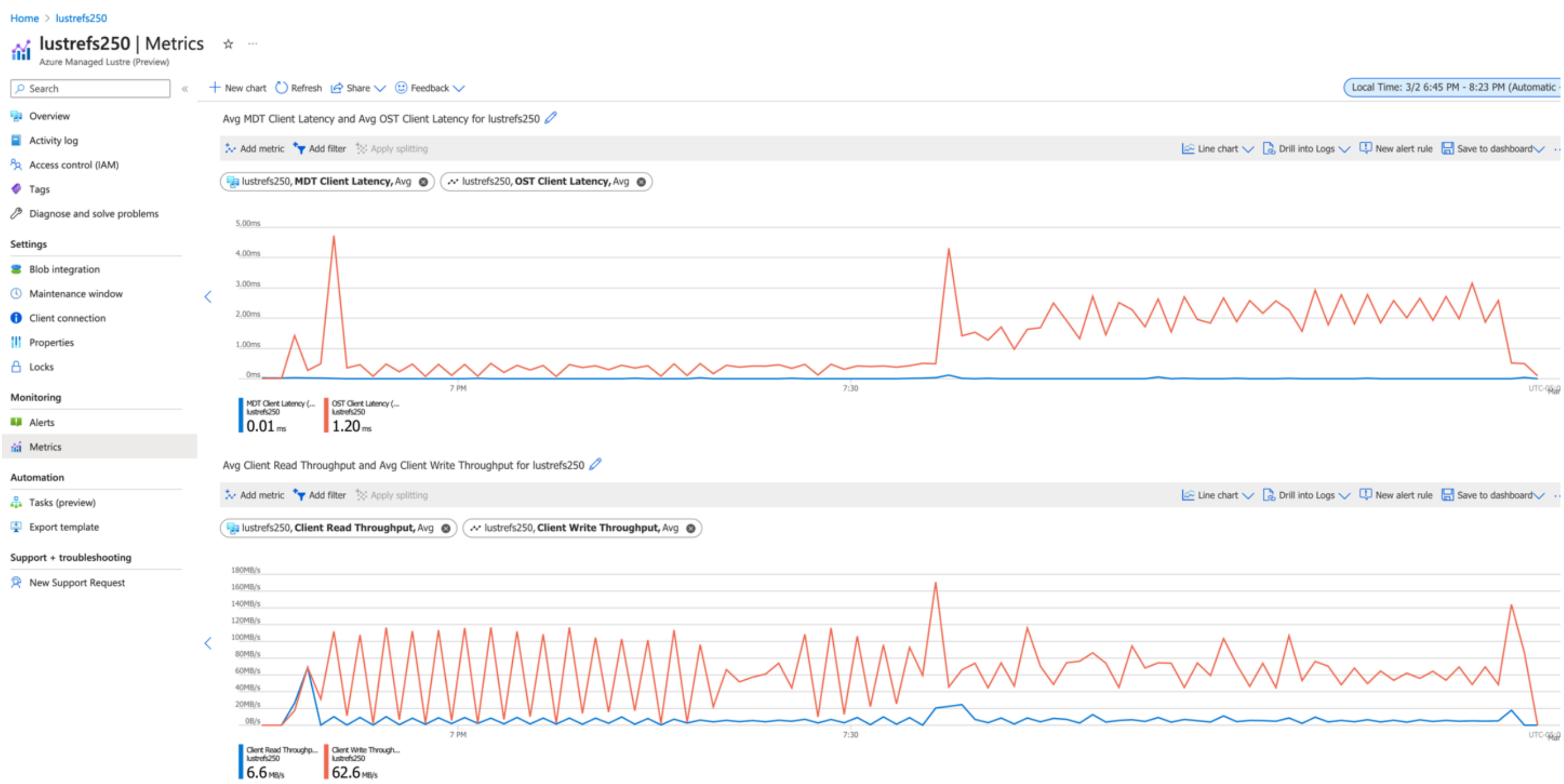

**Figure 19. Azure Monitor™ Cloud Metric latency and throughput measurement of data read, and data write for the cloud benchmark suite using 96 cores on CycleCloud using the Lustre filesystem for full output (all variables, all layers) in CONC file for the cloud benchmark suite (CBS_full).**

### 4.4 ARM MAP™ Code Profiler Analysis for Azure CycleCloud and AWS ParallelCluster

As mentioned before, examining the main log and associated timing plots does not precisely capture the time spent in I/O tasks. The use of a code profiler such as the ARM MAP™ Profiler (ARM Ltd., 2022) can provide better insights on the model components, along with more detailed capture of the I/O tasks by different code routines. Results from applying the ARM MAP™ code profiler on a single day benchmark simulation using the limited I/O benchmark for each CSP cluster and storage offering are presented in Figure 20. We can clearly see a very similar behavior in both systems with the model using a Lustre filesystem. For ***/shared***, AWS's EBS solution is performing much better, with less time spent in I/O that allows speeds closer to physical, "bare-metal" server equivalents. For Azure, the ***/shared*** volume has proportionately higher time spent in I/O percentage and lower percentage spent on computation. In earlier tests without a profiler, the Azure NetApp Filesystem (ANF) solution provided better I/O performance but due to considerably higher cost due to a minimum file size of 4TB we chose not to include an ANF setup in the profiler tests. The performance improvements going from ***/shared*** to ***/data*** and Lustre on Azure's CycleCloud are also demonstrated by comparing Figure 21a and Figure 21b, where orange indicates the I/O portions that get reduced as we utilize faster storage options.

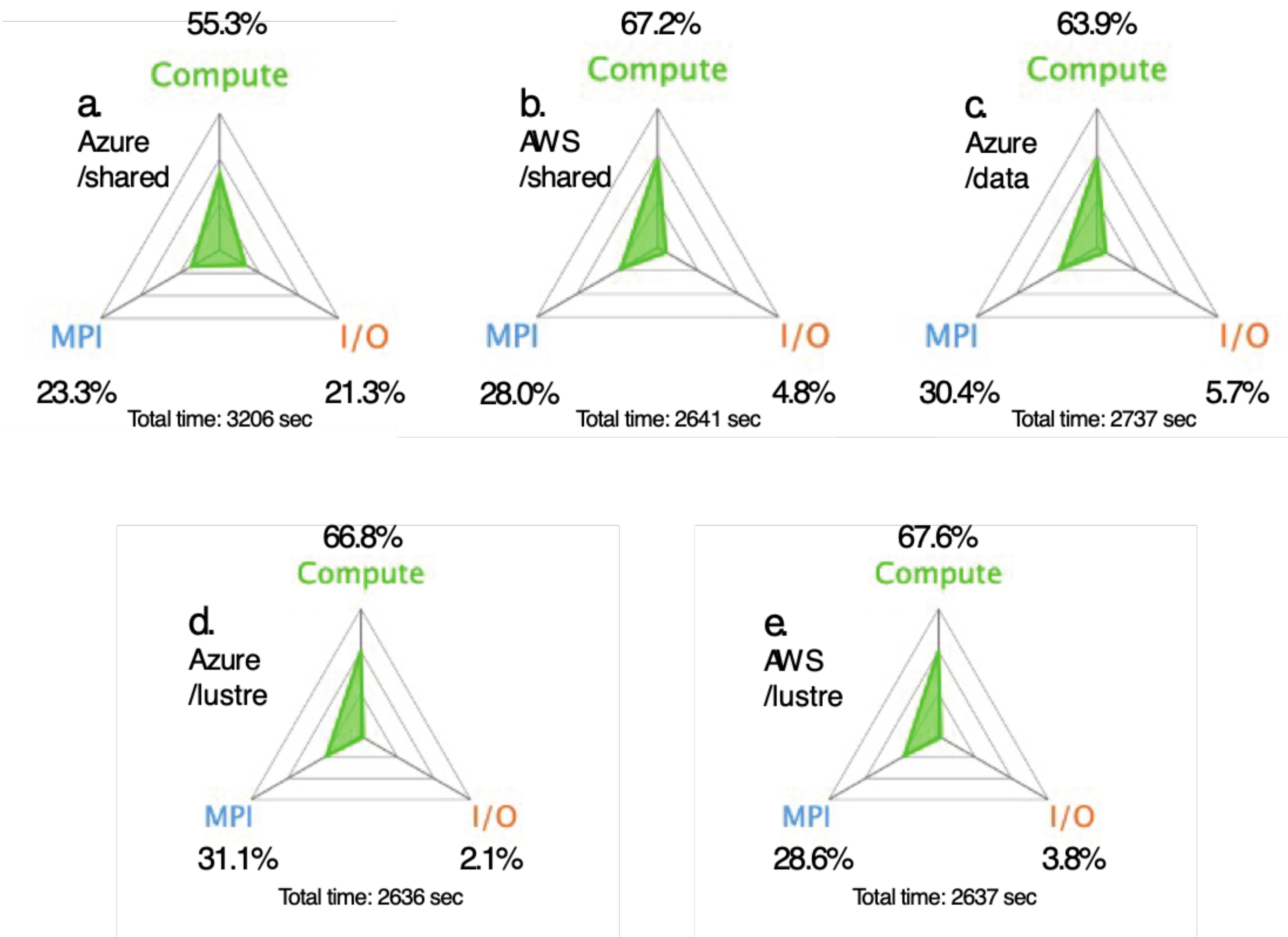

460

**Figure 20: ARM MAP™ Code profiler performance analysis for the first day of the cloud benchmark suite (CBS_limited) on AWS's ParallelCluster and Azure's CycleCloud using different storage options (a. Azure *\/shared*, b. AWS */shared*, c. Azure */data*, d. Azure */lustre*, and e. AWS */lustre*).**

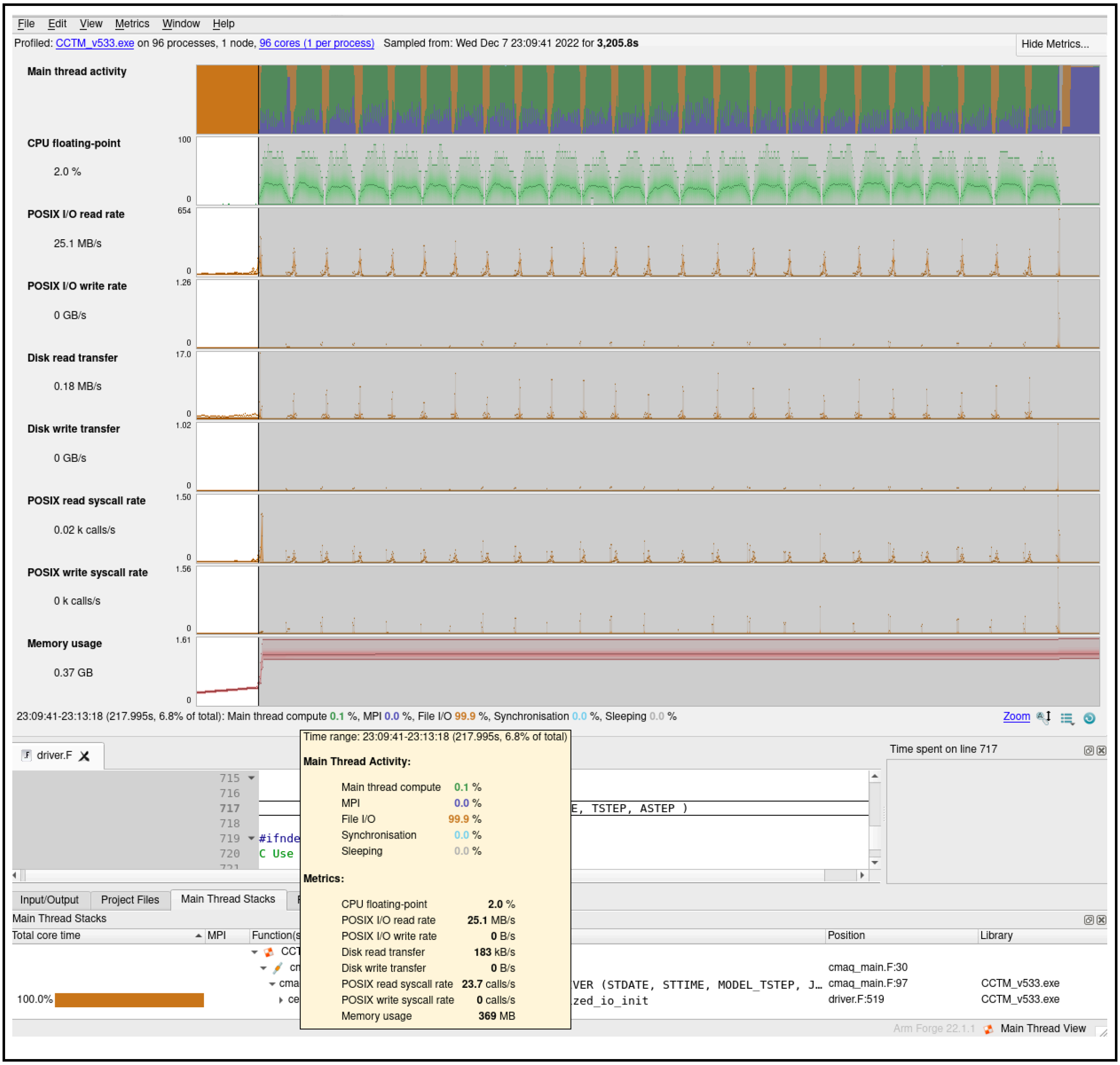

**Figure 21a: ARM MAP™ Code profiler single day performance analysis for Azure's CycleCloud using the NFS /shared filesystem.**

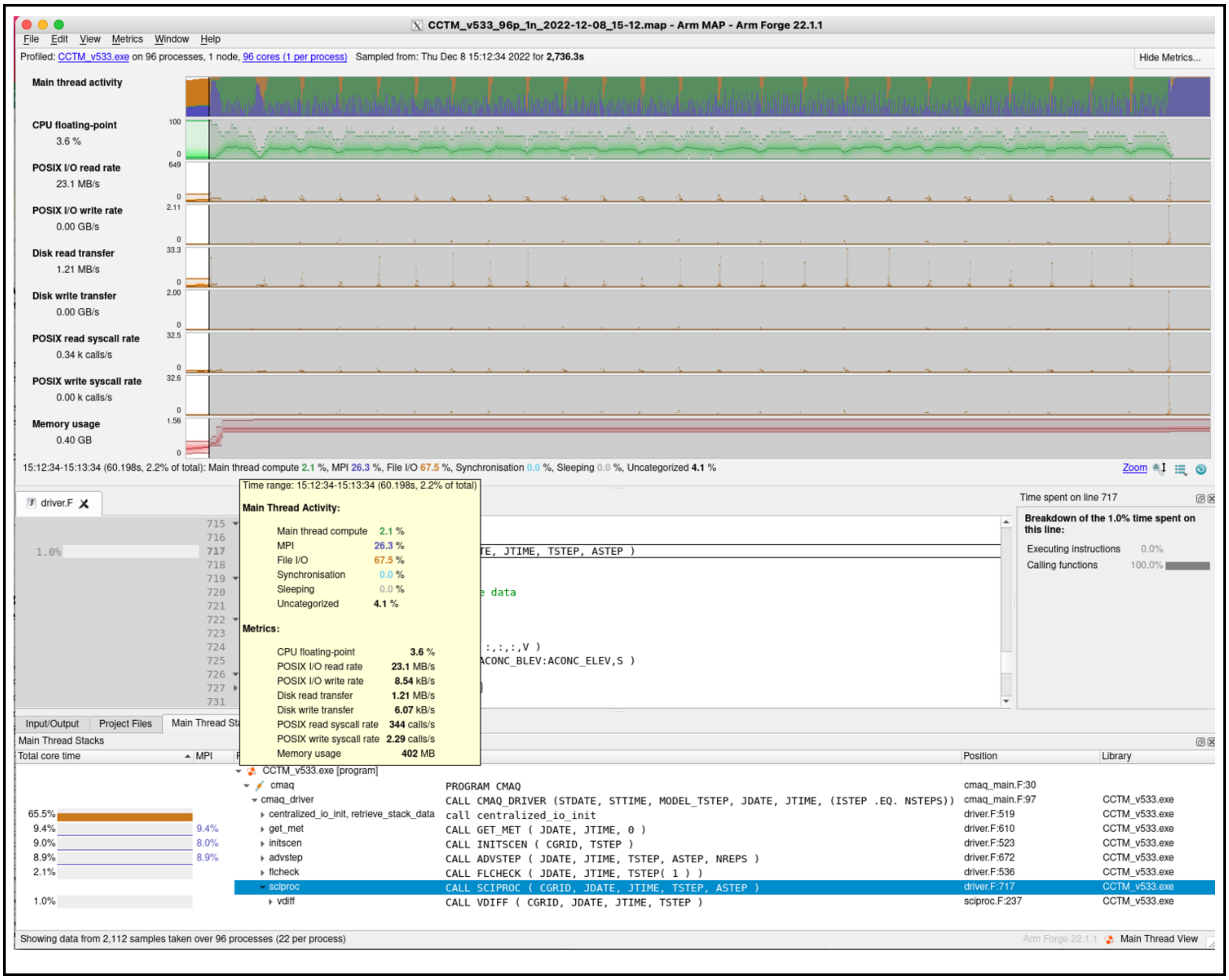

**Figure 21b: ARM MAP™ Code profiler single day performance analysis for Azure's CycleCloud using the /data filesystem.**

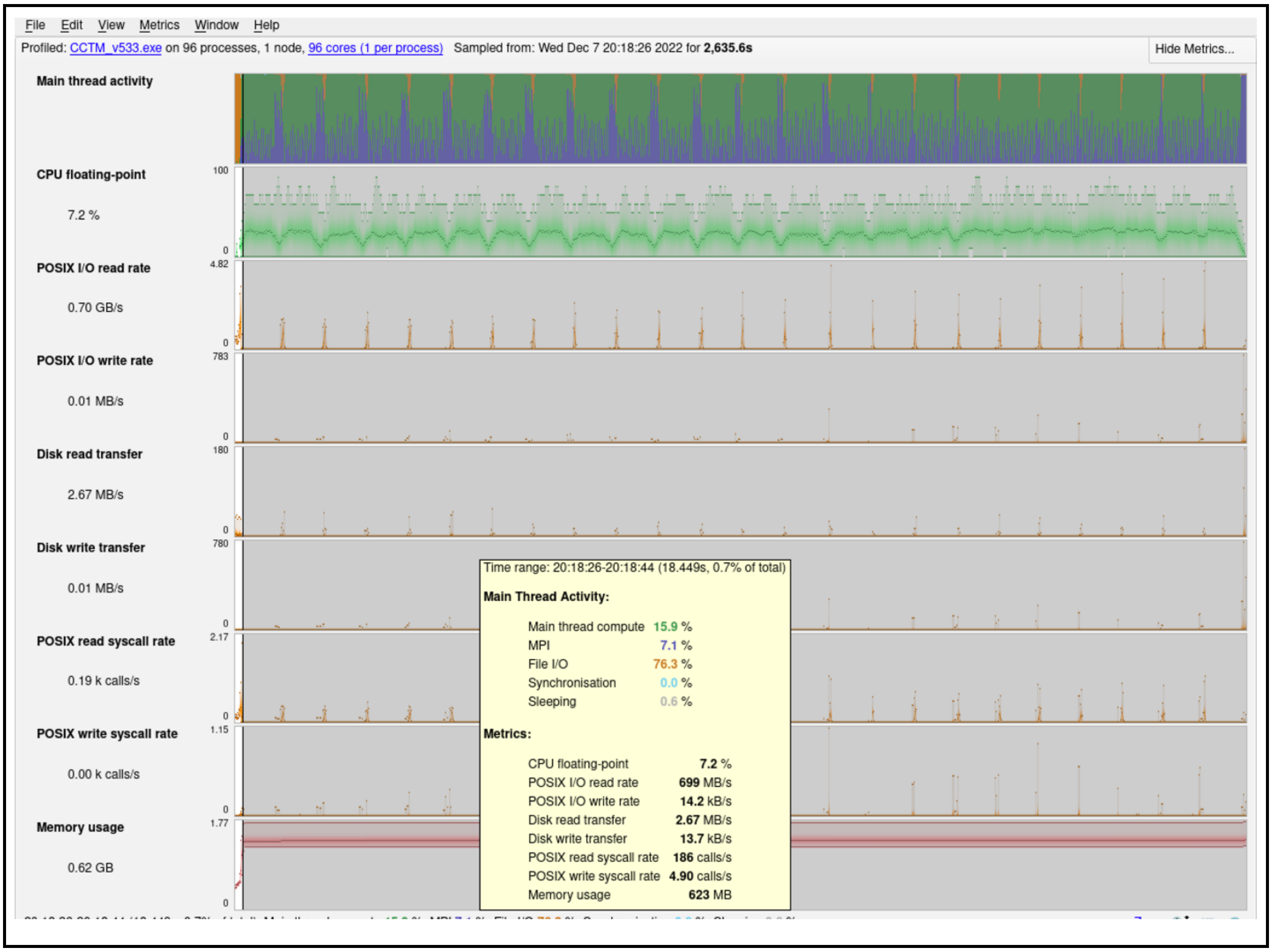

**Figure 21c: ARM MAP™ Code profiler single day performance analysis for Azure's CycleCloud using the Azure-managed *Lustre* filesystem.**

### 4.5 Cost Analysis of Compute Nodes

Table 4 shows comparison of the compute only costs associated with an annual simulation based on the cloud benchmark suite with limited output file options. The setup was based on 2-node cluster setups for both CSP's and the option of spot-pricing that was only available for Microsoft's Azure. It should be noted that spot instances come with an unexpected termination risk that the user should be aware of when designing their implementation. The compute node hpc6a.48xlarge is not provided as a spot instance, but the on-demand price is significantly discounted (60%). However, Amazon does offer spot prices for other compute nodes. This analysis used on-demand pricing options to uniformly evaluate both systems. Users will need to implement code to check-point and recover from a simulation termination if they choose to use spot-pricing or be willing to restart simulations if spot instances are terminated.

**Table 4: Comparison of the compute costs for performing an annual simulation based on the cloud benchmark suite (CBS_limited) on 2-node clusters with on-demand and spot pricing tiers.** Note that these costs are indicative and do not include any other components of the cluster (i.e., storage, head node, etc.)

| Compute Node | Cores | Nodes | Pricing | Cost per node ($) | CBS Wall Time (hour) | Extrapolated Annual Cost | Days to Complete Annual Simulation of CBS |
|---|---|---|---|---|---|---|---|
| HBv3-120 | 192 | 2 | On-demand | 3.6/hour | 0.767 | $1,007 | 5.83 |
| HBv3-120 | 192 | 2 | spot | 1.4/hour | 0.767 | $392 | 5.83 |
| hpc6a.48xl | 192 | 2 | On-demand | 2.88/hour | 0.839 | $883 | 6.4 |

## 5 Discussion

Previous efforts of bringing CMAQ to the cloud demonstrated the potential of packaging the model along with other components as a standalone service optimized for small benchmark domains and low-cost VMs (Zhang et al., 2019). Currently, running complex, computationally demanding models on the cloud presents new options for optimizing workflow, performance, and cost with unprecedented HPC infrastructure. A major implication is a fast deployment of such infrastructures with precompiled software snapshots and preloaded data that are easy to configure and customize according to the user needs. This work evaluates how to run CMAQ on two CSPs and illustrates several issues that should be considered for building HPC clusters on the cloud. We observed that, despite their differences, both AWS and Azure performed similarly, and in accordance with onsite HPC implementations used in earlier phases of this work. There are, however, CSP-specific parameterizations and offerings that may result in more cost-effective solutions for the current CONUS benchmark suite or for more demanding CMAQ applications such as running at higher horizontal grid resolution, e.g., 1-km x 1-km, which is ideal for studying air pollution at local-scales. A very important parameter involves the different storage options as they relate to end user needs.

Figure 6 shows best performance on NVMe drives, which are only available on single virtual machines. For HPC clusters, efficient scaling was achieved by implementing process pinning which resulted in improved performance on /shared, /data and /lustre using Azure's CycleCloud, with the best performance being on Lustre. The effect of process pinning resulted in improved performance on both EBS and Lustre on AWS ParallelCluster. To fully utilize the potential of HPC solutions, the Lustre storage option is advisable for the Azure ecosystem. For AWS, EBS offers a possible, cost-effective alternative to Lustre that may require a complete input and output solution (copying data to/from the S3 Bucket) in the workflow if the ParallelCluster is configured to have the EBS volumes deleted when the cluster is terminated. Scaling performance was improved when both the code and the data files reside on /lustre and may also improve if both the code and the I/O is on local storage (/nvme) or Azure File Share (/data). A key issue that is brought up in the results section is the model scalability which exhibits a diminishing return as the systems are scaled out across more cores and nodes. In general, scalability depends on the domain characteristics (domain size, resolution) and the hardware. Domain decomposition can significantly reduce performance when domain is highly decomposed, i.e., few grid points per core. While our results focus on a fixed size Cloud Benchmark Suite, we expect improved scalability with a higher-resolution domain, as it would increase the workload per core. Compared to typical WRF benchmarks where a 2.5-km x 2.5-km resolution CONUS domain (63 million grid points) is considered a typical case that scales well up to a few hundred cores, the 12-km x 12-km CMAQ CBS (97 thousand grid points) is comparatively too small of a problem to scale similarly. This can be an important focus area when looking at future CMAQ code optimizations and can be explored with more demanding model configurations.

The online tutorials and documentation include additional and ongoing work such as: 1) running the benchmark using new releases of virtual machines (Elastic Compute Cloud (EC2) Instances or Microsoft Azure VMs), 2) building with the EPYC processor including Standard_HB176rs_v4 on Azure Cycle Cloud and on new releases of the Arm-based AWS Graviton-3 processor using c7g.16xlarge on ParallelCluster, 3) running the benchmark using a parallel I/O API implementation and other efficiency improvements in the source code that can be compiler and processor architecture specific. Impact on performance needs to be examined after each model release and for each model configuration and input platform data which varies by year and model parameterization. Future work will also be focused on providing tutorials and installation guides for other tools involved in supporting the CMAQ workflow, including the SMOKE modeling system to provide CMAQ input data and the Atmospheric Model Evaluation Tool (AMET) for comparing model outputs against ground level measurements and satellite observations (Appel et al., 2011).

On the question of conducting simulations on CSPs versus an on-premise option that may be available, the two key areas driving such a decision will be the broader context of the on-premise option, and the specifics of the work to be accomplished. Given no context on these two categories of attributes, the default recommendation is to use a Cloud Service Provider. For on-premise to be a viable option, there generally must be access to some existing infrastructure and services supporting computational research and data storage. In some cases, organizational investments to support computational and data needs

may effectively subsidize the cost of an on-premise solution. These scenarios often come with higher levels of service regarding maintenance and operation, but with the price of constraints and limitations of those services as compared to the freedom of building one's own solution within a CSP. Given a viable on-premise option, it may be reasonable to utilize both on-premise and CSP services if the unique benefits of each of these options are sufficient, and data locality issues do not overwhelm the viability of a hybrid approach. While obtaining exact cost estimates for a traditional HPC system can be challenging, our findings support the conclusion by Zhuang et al. (2020) who found that atmospheric modeling in the cloud was cost competitive and a viable alternative to more traditional HPC systems.

This work provides reproducible workflows to facilitate provisioning of HPC clusters on the cloud, setting up and running CMAQ and provides detailed information on using performance analysis tools and profilers to identify the bottlenecks in achieving efficient use of the HPC systems. The very nature of cloud implementations comes with continuous advancements in compute, memory, and storage resources that together with the resources developed under this work can be used to optimize for the problem size allowing to scale up and run more efficiently on HPC systems every time a component limitation or a change in the configuration is justified. An example is AWS modification to the EPYC chip to create compute nodes with hypervisor on a different Nitro chip, and isolating the CPUs on the compute node for user's application from cache management tasks performed by the hypervisor on the Nitro chip. Our HPC in the cloud paradigm allows researchers to improve their code, workflow and to access a menu of specialized HPC resources offered by cloud computing vendors resulting in faster time to solution. Broad use of these tutorials by the CMAS Community will facilitate use of best practices for HPC cloud provisioning, increase cross-institution collaborations, and improve efficiency in code development and deployment cycles.

**Code and data availability**

The code for CMAQ is available and referenced here:
https://github.com/USEPA/CMAQ/
https://doi.org/10.5281/zenodo.5213949
The code for the AWS cloud implementation is available and referenced here:
https://github.com/CMASCenter/pcluster-cmaq/tree/CMAQv5.3.3
https://doi.org/10.5281/zenodo.10696908
The code for the Azure cloud implementation is available and referenced here:
https://github.com/CMASCenter/cyclecloud-cmaq/tree/CMAQv5.3.3
https://doi.org/10.5281/zenodo.10696804
Data inputs for the Benchmark Suite are available and referenced here:
https://registry.opendata.aws/cmas-data-warehouse/
https://doi.org/10.15139/S3/CFU9UL
Tutorials with instructions on running CMAQ version 5.3.3 and above on the cloud are available through:

https://cyclecloud-cmaq.readthedocs.io/en/cmaqv5.3.3/

https://pcluster-cmaq.readthedocs.io/en/cmaqv5.3.3/

**Supplement**

The supplement related to this article is available online as an Appendix below.

**Author contributions**

CE and LA share joint first authorship of the study. CE and LA designed the study, implemented the parameterizations and setups, performed the simulations, analyzed the results, and drafted the original draft of the paper. CC and DW contributed to various model optimizations and troubleshooting model performance. RZ, MR and JM contributed to provisioning Microsoft Azure resources for the modeling, troubleshooting modeling issues on Azure and comparisons against AWS. KF, FS and SF contributed to multiple model configuration tweaks, refining parameterizations, troubleshooting and interpretation of results. SA conceived the study and provided project oversight and guidance. All authors contributed to comments and revisions of the original draft.

**Acknowledgements**

We acknowledge Amazons' Open Data Program for hosting the datasets and Microsoft for providing cloud computing credits for the work performed in this study. Steve Roach of Microsoft provided guidance on configuring the CycleCloud, using pinning to improve performance with the HB120v3 compute nodes, and access to the Lustre filesystem. Rafa Salas of Microsoft provided access to cloud experts and facilitated a grant to UNC with computing credits. Timothy Brown and Tommy Johnston of AWS provided guidance on configuring ParallelCluster using HPC6a compute nodes. Chris Stoner of AWS provided guidance on enrollment in the AWS Open Data Sponsorship Program and technical assistance with the Handbook for Data Providers that was used to create the CMAS Data Warehouse, and further making the datasets searchable on the AWS Data Exchange. We also acknowledge the contributions of Zac Adelman (LADCO) as co-lead of the Air Quality Modeling in the Cloud Computing Workgroup Co-Leader with Steve Fine and Fahim Sidi (EPA), who helped guide our modeling effort on the cloud.

**Disclaimer.**

This paper has been subjected to an EPA review and was approved for publication. The views expressed here are those of the authors and do not necessarily reflect the views and policies of the US Environmental Protection Agency (EPA), or any of the CSPs named here.

**Financial support**

The U.S. EPA, through its Office of Research and Development, partially funded and collaborated in the research described here under EP-W-16-014 to UNC Chapel Hill. Microsoft provided credits for performing the necessary benchmark simulations on Azure.

**Competing interests**

UNC through its Information Technology Services Research Computing division received cloud computing credits from Microsoft that were used for the work performed here, and received sponsorship from AWS for enrollment in the Open Data Program that are used to host the datasets mentioned here.

**Appendix**

A growing number of companies ranging from large enterprises such as Amazon, Microsoft, and Google, to a spectrum of cloud-focused computer firms, have a strong presence with evolving portfolios in what is broadly defined as "public cloud infrastructure". Typically, cloud computing is provided through at least four types of services summarized in Figure A1: Infrastructure as a service (IaaS), Platform as a Service (PaaS), Software as a service (SaaS), and Data as a Service (DaaS) (Mell and Grance, 2011; Chang et al., 2010; Yuan, 2016). IaaS products (Amazon Web Services, Microsoft Azure, Google Cloud, etc.) allow organizations and end-users to manage their system resources (i.e., servers, network, data storage) on the cloud. PaaS products (Windows Azure, Google app engine, etc.) allow businesses and developers to host, build, and deploy consumer-facing apps. The most important contrast between IaaS and PaaS is that IaaS gives users more administrative control over operating systems, while PaaS gives consumers the ease of use of provided applications, but limits access to choices about the operating system. SaaS products are among the most popular cloud computing services (Microsoft 365, Google docs, Adobe cloud, etc.) offering out-of-the-box, simple solutions that usually target common end users and operate under a subscription model. DaaS, the least well-defined type of service describes cloud-based software tools used for working with data, such as managing data in a data warehouse entity, processing, or analyzing with software tools, typically enabled by SaaS technologies under a subscription model. HPC applications require administrative access to networking, hardware, and storage configurations, and therefore need infrastructure as a service (IaaS) level of control that are provided by cluster services.

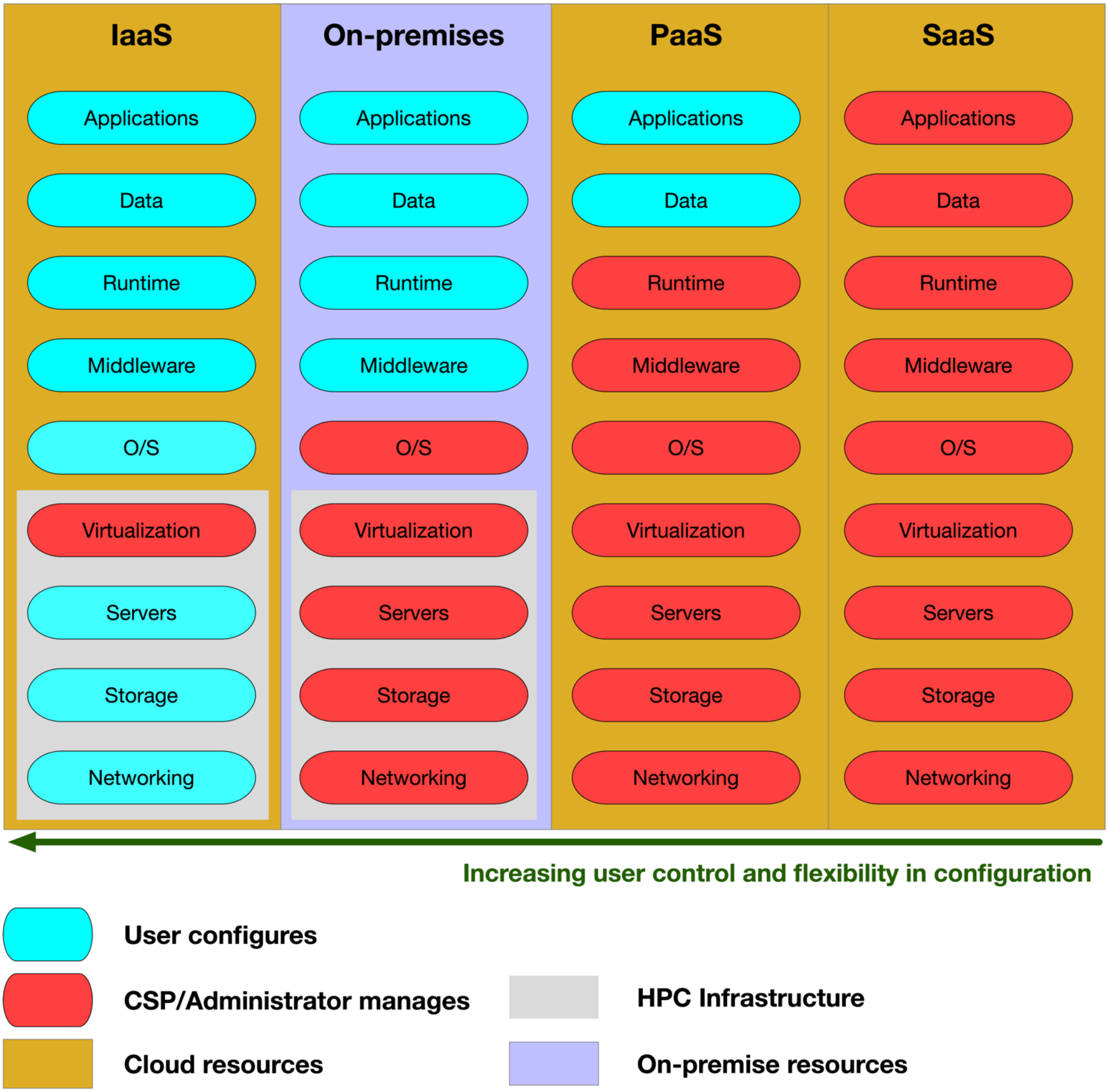

640 Figure A1: Architecture of on-site and cloud-based services demonstrating the degree of end-user control.

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
