# Peer review of "Enabling High Performance Cloud Computing for the Community Multiscale Air Quality Model (CMAQ) version 5.3.3: Performance Evaluation and Benefits for the User Community"

_EGUsphere, 2023_

## Author Comment (AC1)

**Response to Reviewer 1:**

*We would like to thank reviewer 1 for the valuable comments and suggestions. A revised version of the manuscript that addresses the review comments is provided along with point-by-point responses below.*

RC1: Anonymous Referee #1, 21 Apr 2024  The manuscript introduces the adaptation and optimization of the CMAQ 5.3.3 for high-performance cloud computing environments. This paper fits the scope of GMD and can serve as a detailed reference to showcase how the CMAQ model can enhance computational efficiency and accessibility for diverse modeling tasks. Here are some minor suggestions that could be addressed to further improve the paper.

**Response:** *We thank the reviewer for the positive and encouraging comments on our manuscript. Please see detailed responses below.*

Line 115-125: How about illustrating the CMAQ workflow using a figure? It would help readers better understand how CMAQ works.

**Response:** *We have incorporated the reviewer's suggestion in new **Figure 3**.*

Line 130-150: This is lengthy and somewhat difficult to follow. Please break it into multiple paragraphs to enhance readability.

**Response:** *We have restructured th**e** paragraph as suggested  to enhance readability**. The new content is in lines 130-165.*

Line 160: Figure 1 only shows the rectangle of CONUS but lacks grid representation. I suggest exemplifying the grids over an area of interest with a zoom-in minimap.

**Response:** *We have added a new figure depicting a subdomain of CONUS over NY with grid lines **(Figure 2).***

Line 165-290: Section 3 offers valuable insights into CMAQ deployment from an engineering perspective. However, to align more closely with the scientific paper, consider pivoting towards system or experiment design to elucidate the methodology behind this work, while relocating detailed technical tutorials to an appendix.

**Response:** *The additional figures 2, 3, 13 help better elucidate the methodology and code surrounding the experimental design, by illustrating examples of existing functionality (DESID module) for the example of the CONUS cloud benchmark suite. The links to the tutorials are provided to allow the users to not only reproduce the results of the suite, but also define the experimental design for their individual needs, that can often expose unique limitations and require additional benchmarking and optimization pathways (i.e.,*

*cpu/memory/storage/networking). Given this, the links could be more useful to end-users if made directly available.*

Line 335-410: How about combining Figure 6/7, 8/9/10, 11/12/13/14/15/16? It is a little bit hard for readers to compare the results across multiple figures.

**Response:** *We have combined these individual figures as suggested in new Figures 8 - 12.*

Line 495-560: The current discussion could be streamlined and organized into subtopics, such as the strengths of the proposed cloud-based implementation, scalability/reusability, limitations, and future research recommendations. Meanwhile, a conclusion section is recommended to summarize the research findings from this work.

**Response:** *We have restructured and rewritten the discussion section into 5 subsections to consider the reviewer's suggestion. In addition, we also incorporated a conclusion section at the end of the manuscript to summarize the research findings.*

---

## Author Comment (AC2)

**Response to Reviewer 2:**

*We would like to thank reviewer 2 for the valuable comments and suggestions. A revised version of the manuscript that addresses the review comments is provided along with point-by-point responses below.*

RC2: Anonymous Referee #2, 23 Apr 2024
This manuscript presents a research effort to enable CMAQ modeling and data analysis on high performance cloud computing. CMAQ is a popular air quality model and has been widely applied for numerous regulation and research purposes. The application of CMAQ however, is somehow still limited since it requires preparing all inputs and run scripts on one single server. Enable CMAQ on cloud server would make it more convenient to run the model and would probably promote its applications to a broader community. The study is really a worth of efforts. The manuscript provided clear descriptions of the flow chart and sufficient details of each section of the modeling system, and also demonstrated the changes in cost and efficiency clearly. Therefore, I would recommend it to be accepted for publication, if the following minor comments could be properly addressed.

**Response:** *We thank the reviewer for the positive and encouraging comments on our manuscript. Please see detailed responses below.*

comment#1: Computational efficiency for traditional CMAQ is not linearly increasing with more CPU cores and data I/O is a big reason. But for cloud-based CMAQ it seems horizontal advection is most time consuming, which is a little surprising. Please provide a brief discussion regarding this change.

**Response:** *The behavior described by the simulations performed on cloud resources followed a pattern similar to traditional HPC setting; HADV is an important component of the model data flow that can contribute significantly when more compute resources are provided to coarse domain setups. A quote from literature for GEOS-CHEM (a global-scale air quality model) states a similar conclusion: "The reason is that large core counts at coarse resolutions result in excessive internode communication for advection relative to computation within the node (Martin et al., 2022)". Research for a larger domain such as the 12US1 shows similar performance behavior for HADV. (Delic, 2020). The following sentence was added to the text for clarity in lines 336-339:* "Horizontal advection is the most time consuming of the processes within CMAQ. This is most likely due to communication between processors during advection which requires information from neighboring cells to calculate advective fluxes. This is domain dependent and there can be domains where the computational demand is very large (e.g., applications like MPAS) that one may not see this trend till one uses thousands of cores. In short, more cores results in less work per core but more time is needed for each core to communicate with each other."

comment#2: I guess the current cloud version doesn't support two-way mode WRF-CMAQ, please clarify if it is correct. Also, does it support online modules for MEGAN and dust emission?

**Response:** *CMAQ calculates emissions rates online for several sources including biogenics, wind-blown dust, sea spray, and lightning NO. In CMAQv5.3.3 the online biogenics are calculated using the BEIS model. The option to use MEGAN online biogenic emissions was introduced in CMAQv5.4. Users can follow the tutorial and use a more recent version of CMAQ to enable the MEGAN option in their cloud simulations.*

*WRF-CMAQ is not a part of the current manuscript as it adds additional complexity to the workflow. We now mention WRF-CMAQ in the discussion section on future work. In addition, users can follow the tutorial available here that uses a smaller 12NE3 domain. However, it is not ideal for benchmarking on the cloud due to the small domain size.*
*https://github.com/USEPA/CMAQ/blob/main/DOCS/Users_Guide/Tutorials/CMAQ_UG_tutorial_WRF-CMAQ_Benchmark.md*

comment#3: Fig.3 and Fig.4 is not mentioned in the main text. It's necessary to briefly explain the flowchart although the figure is quite self-explained.

**Response:** *Figures 5 and 6 (formerly figures 3 and 4 respectively) are now properly referenced in the main text, under "Cloud HPC Configuration Summary".*

comment#4: Fig.8 and Fig.9: It's interesting to notice that pinning on Azure speeds up vertical diffusion but on AWS slows it, please provide a discussion to briefly explain the difference.

**Response:** *It is unclear why this occurred, and we thank the reviewer for pointing this. . Different CSPs use different hardware versions (EPYC processor versions), with different hypervisor and SLURM implementations, that result in different number of cores available per node and a different approach to pinning instructions. In the case of AWS, the implementation resulted in fewer cores/node available to the user (96 vs. 128), but the implementation was automated and enabled without the additional fine-tuning steps that were required for Azure (manually enabling pinning options for AWS had minimal impact on compute timings). The idea behind including pinning results is to make end-users aware of newly available hardware-software optimization areas beyond traditional core/memory size, and its dependence on the CPU architecture (along with revisions/updates for individual CSP implementations).*

comment#5: It's important to notice that only a few variables are saved to 1-layer conc file during the test in section 4.2.2, while in real application the variables and layers may be much more and larger. Please provide a brief discussion to justify if the test runs shown in this study are representative for typical CMAQ applications.

**Response:** *We performed additional benchmark runs and added them to Figure 12. CMAQ timings still show speed-up when going from 1 node to 3 nodes, even with all layers, all variables output to CONC. Many of the QA/QC and/or post processing steps use ACONC output, which contains 1 layer, all variables. Comparisons against observational data and other analyses such as assessing health effects of air quality, etc. is typically performed using data from the first layer. It is only if the scientific study required nesting to a finer domain, that all*

*layers, all variables would need to be saved to the CONC file for subsequently extracting boundary conditions.*

comment#6: Fig.18, Fig.19, and Fig.21: Showing screen print is straightforward but a little improper for journal publication, it's better to summarize the important numbers into a concise figure or table.

**Response:** *We have moved the Tools for Storage Performance Monitoring section and figures to the supplemental information document (see Figures A2 to A7) to enhance the manuscript readability.*

*However, the value in sharing these images in the paper is that these types of tools are not available for on-premise compute servers. The CSPs make these tools available to help users understand the impact of bandwidth, CPU and I/O speed on code performance, and we would like to share these resources with the user community.*

**References for the response to comment 1 from reviewer 2:**

*Martin, R. V., Eastham, S. D., Bindle, L., Lundgren, E. W., Clune, T. L., Keller, C. A., Downs, W., Zhang, D., Lucchesi, R. A., Sulprizio, M. P., Yantosca, R. M., Li, Y., Estrada, L., Putman, W. M., Auer, B. M., Trayanov, A. L., Pawson, S., and Jacob, D. J.: Improved advection, resolution, performance, and community access in the new generation (version 13) of the high-performance GEOS-Chem global atmospheric chemistry model (GCHP), Geosci. Model Dev., 15, 8731–8748, https://doi.org/10.5194/gmd-15-8731-2022, 2022.*

*Delic, 2020 https://www.cmascenter.org/conference/2020/abstracts/CMAS-CMAQ-2020-delic.pdf*